# Transposable elements orchestrate subgenome-convergent and -divergent transcription in common wheat

Yuyun Zhang[1,2,3,11], Zijuan Li[1,2,3,11], Jinyi Liu[1,2,11], Yu'e Zhang [2,4,11], Luhuan Ye[1,2,11], Yuan Peng[1,2,5], Haoyu Wang[1,6], Huishan Diao[3], Yu Ma[1,2,5], Meiyue Wang[3], Yilin Xie[1,2], Tengfei Tang[1,6], Yili Zhuang[1,2], Wan Teng[2,4], Yiping Tong[2,4], Wenli Zhang [7], Zhaobo Lang[1,2,5,8] ✉, Yongbiao Xue [2,4,9,10] ✉ & Yijing Zhang [3] ✉

The success of common wheat as a global staple crop was largely attributed to its genomic diversity and redundancy due to the merge of different genomes, giving rise to the major question how subgenome-divergent and -convergent transcription is mediated and harmonized in a single cell. Here, we create a catalog of genome-wide transcription factor-binding sites (TFBSs) to assemble a common wheat regulatory network on an unprecedented scale. A significant proportion of subgenome-divergent TFBSs are derived from differential expansions of particular transposable elements (TEs) in diploid progenitors, which contribute to subgenome-divergent transcription. Whereas subgenome-convergent transcription is associated with balanced TF binding at loci derived from TE expansions before diploid divergence. These TFBSs have retained in parallel during evolution of each diploid, despite extensive unbalanced turnover of the flanking TEs. Thus, the differential evolutionary selection of paleo- and neo-TEs contribute to subgenome-convergent and -divergent regulation in common wheat, highlighting the influence of TE repertory plasticity on transcriptional plasticity in polyploid.

Polyploidy is a major factor driving plant evolution and speciation, which is particularly prevalent in plants[1–4]. Polyploids demonstrated increased adaptability and plasticity compared with their progenitors in evolution. This has been attributed to the diversity and synergy among different subgenomes[3,5–7], raising the major question about how subgenome-divergent and -convergent regulation is achieved and harmonized in polyploids.

Common wheat (*Triticum aestivum*, 6x = AABBDD) contains three sets of different genomes which underwent diverge-and-merge speciation events (Fig. 1a)[8]. The diploid progenitors of the three subgenomes diverged from a common ancestor about five million years ago[9], resulting in highly diversified intergenic regions with a near-complete turnover of transposable elements (TEs)[10]. Two successive polyploidization events occurred ~0.8 million years ago and 9000 years ago, which retained the genomic diversity of these diploid progenitors[7]. Subgenome diversity and the buffering effects of polyploidy were proposed to be major factors that contributed to the high plasticity of common wheat[5,7]. Further domestication lead to the development of common wheat as a staple crop cultivated worldwide.

The large intergenic regions of common wheat harbor abundant regulatory elements (REs) encoding regulatory information that determines the temporal and spatial specificity of genes[11–13]. The associated variations affect a wide range of agronomic traits[14–17]. Intergenic variation of REs across subgenomes may help explain the fact that the expression of 30% of wheat homoeologs is unbalanced[12,13,18]. However, it remains unclear how RE diversity across subgenomes is specifically interpreted to dictate subgenome-biased transcription. TEs are a rich source of REs as reported in both

animals[19–21] and plants[22–27]. Near-complete TE turnover was detected in intergenic regions of common wheat across subgenomes. To what extent subgenome-diversified TEs contributed to subgenome-biased transcription is unclear. Furthermore, despite the highly diverse intergenic regions, earlier researches revealed the extensive balanced expression of homoeologs throughout development[18,28], raising an additional question regarding how this evolutionary constraint on transcriptional regulation was achieved. The specific recognition and binding of transcription factors (TFs) to REs is a primary mechanism by which cells interpret genomic features[29]. Elucidating the extent to which TF binding differs across subgenomes as well as the global relationship between TF binding and subgenome variations in REs is critical for addressing the above-mentioned issues.

In this work, we assemble a common wheat regulatory network comprising connections among 189 TFs and 3,714,431 REs, which help enhance the understanding of wheat regulatory mechanisms. By leveraging phylogenetic strategies to study the evolution of the regulatory map, we not only detect lineage-specific TE expansions and exaptations for subgenome-divergent transcriptional regulation, but also track diploid parallel selection on transcription factor-binding sites (TFBSs) derived from ancient TE expansions. Our findings connect the dynamic death and birth of TEs to regulatory evolution in common wheat, demonstrating that the plasticity of TE repertoires potentially influence polyploid plasticity.

## Results

### Genome-wide profiling of TFBSs in common wheat

We cloned 189 TFs from 30 families, of which 107 were highly expressed TFs and 82 were functionally annotated TFs or hub TFs in the co-expression network (Supplementary Data 1). Each clone was verified by full-length cDNA sequencing to confirm a lack of chimeric fragments from homoeologs. Next, a DAP-seq analysis[30] was performed to characterize the genome-wide binding of these TFs, which were classified according to whether the canonical binding motif was de novo identified or enriched in a given TF peak list. This analysis resulted in 45 high-, 47 median-, and 97 low-confidence TF datasets (HC, MC, and LC, respectively) (Fig. 1b and Supplementary Fig. 1). All DAP-seq data and peak files were deposited in GEO database [https://www.ncbi.nlm.nih.gov/geo/query/acc.cgi?acc=GSE192815]. The HC and MC TFs were used for the subsequent analysis. The DAP-seq success rate, represented by the fraction of HC TFs for each TF family, varied among TF families. More specifically, the AP2, MYB, and B3 TF families had high, median, and low success rates, respectively (Fig. 1c). The binding for the TF families with low success rates likely requires co-factors. All data were visualized using a customized genome browser (http://bioinfo.sibs.ac.cn/dap-seq_CS_jbrowse/). Transcription factors from the same family generally had similar binding profiles (Fig. 1d). The binding of homoeologous TFs was largely similar across subgenomes (enlarged heatmap in Fig. 1d), implying that the binding specificity is likely dependent on RE sequences.

The TFBSs are not randomly distributed throughout the genome, with regions containing 42,332 binding sites designated as high-occupancy target (HOT) regions[31,32]. The high regulatory activities of HOT regions were reflected by the relatively high levels of chromatin openness characterized by a DNase I hypersensitive site (DHS) and H3K9ac activity typical of active promoters and enhancers in wheat[11,33] as well as the conservation across four wheat species with different polyploidy levels (see Methods) (Fig. 2a). Additionally, 53% of the HOT regions had sequences in syntenic regions that were conserved in three subgenomes (Fig. 2b). Most of these sequences were in gene-proximal regions (Fig. 2c). By comparing HOT regions with higher-order chromatin structures, we determined that HOT regions were preferentially localized to topologically associating domain (TAD) boundaries (Fig. 2d). Figure 2e presents the genomic features of one subgenome-conserved HOT region. Although the

local chromatin structure varied substantially across subgenomes, HOT regions were still preferentially localized to TAD boundaries. Previous research indicated TADs are formed via promoter–enhancer linkages mediated by co-opted REs[34]. In the current study, the considerable enrichment of HOT regions in TAD boundaries implies that a high TF occupancy may be associated with TAD formation. Alternatively, the chromatin architecture in TAD boundaries may help facilitate TF occupation.

On the basis of this regulatory information, a directed regulatory network was constructed (Supplementary Fig. 2), with the TF-target gene pairs listed in Supplementary Data 2. To demonstrate the functional implication of the binding of TFs, we integrated co-expression profiles derived from 200 transcriptomic datasets, resulting in eight modules with connections among 34 TFs and 8937 genes. To characterize the functions of these modules, we screened for enriched Gene Ontology (GO) terms using GOMAP[35]. The functionally annotated groups are summarized in Fig. 2f and Supplementary Fig. 3. A module comprising TFs and targeted genes potentially involved in photosynthesis is presented in Fig. 2g (zoomed in on the right). The module consists of thoroughly describes TFs and other factors related to the photoperiod and photosynthesis, including Dof, Ppd1, and Elf3[36]. Homoeologous TFs generally have similar target genes. This directed regulatory map allowed us to explore how polyploidy is regulated and the associated effects on evolution.

### Expansion of TFBSs in common wheat

To compare the RE architecture across subgenomes, HC TFBSs were divided according to their sequence conservation among subgenomes (Fig. 3a). Subgenome-homologous regions were detected on the basis of a reciprocal alignment, with syntenic (homoeologous) and non-syntenic regions calculated separately (see Methods). On average, 44% of the TFBSs were localized in subgenome-specific regions (i.e., unalignable to the other two subgenomes), indicating pervasive asymmetric subgenome regulation. To examine the diversity in the functions of the genes regulated by subgenome-convergent and -divergent TFBSs, we searched for the over-represented GOMAP terms associated with genes preferentially containing subgenome-homoeologous and -specific TFBSs, respectively. The most enriched GO term among the genes with homoeologous TFBSs was membrane architecture (Fig. 3b), whereas genes with subgenome-divergent TFBSs were mostly related to defense, with sequences that varied among wheat species (Fig. 3c). Thus, subgenome-divergent environmental adaptation is likely mediated by subgenome-divergent regulatory circuits.

We next examined the origin of subgenome-divergent TF binding. For each TF, the TFBSs localized to subgenome-specific regions were included in a pair-wise sequence comparison for each subgenome to identify TFBS pairs with similar sequences. A Circos plot (Fig. 3d) connecting bHLH-1A-1-binding sites with highly similar sequences within each subgenome was constructed. These sites were revealed to be much more abundant in wheat than in Arabidopsis thaliana. The pair-wise sequence distance distributions of TFBSs within subgenome-specific regions were determined for all TFs (Fig. 3e and Supplementary Fig. 4). Clearly, almost all TFBS regions underwent at least one expansion event during evolution, as reflected by the apparent peak(s) indicating the sequence similarity among a number of TFBSs. This is in contrast with the results of other thoroughly investigated model plants, including A. thaliana and Oryza sativa (Fig. 3d, e and Supplementary Fig. 4).

### Different TE families contribute to subgenome-specific TFBSs

More than 80% of the TFBSs with high pair-wise sequence similarities were detected in transposable elements (TEs) (Supplementary Fig. 5), whereas <40% of the TFBSs without high pair-wise sequence similarities were localized in TEs. Given the high abundance of TEs and

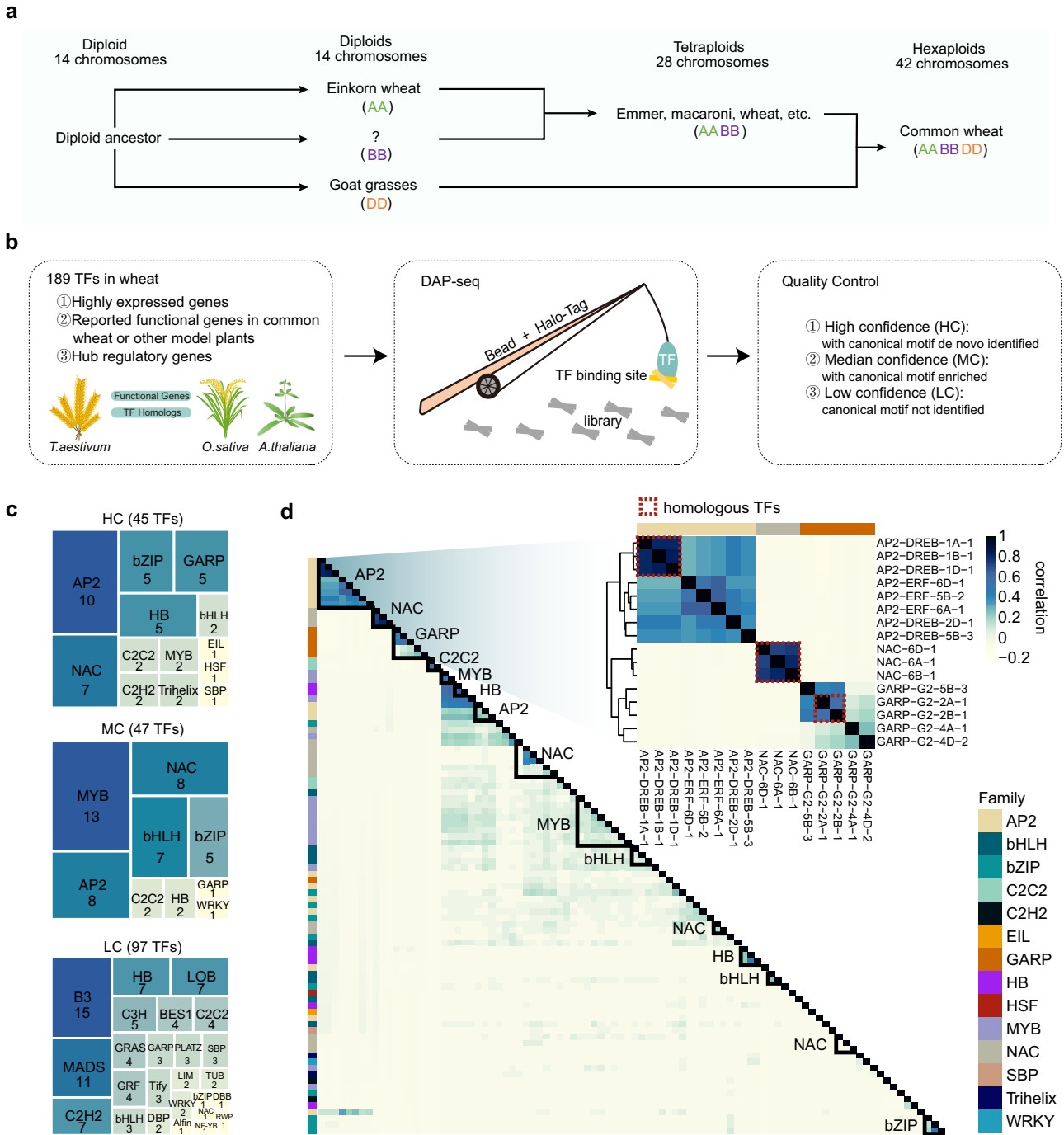

**Fig. 1 | Genome-wide binding of wheat transcription factors. a** Evolution of domesticated wheats (modified from ref. 8). **b** Schematic of the experimental design and filtering steps. **c** Tree map illustrating the fraction of transcription factor (TF) families defined as high-confidence (HC; canonical motif de novo identified from the TF peak list), median-confidence (MC; canonical motif enriched in the TF peak list), and low-confidence (LC; canonical motif not enriched in the TF peak list) families. **d** Clustering of TF binding correlations according to the occurrence of DAP-seq peaks. The genome is divided into consecutive 2 kb bins. For each TF, bins are binarized, indicating the presence or absence of TFBSs, which are used to calculate the Pearson correlation coefficient. The heatmap of the clustering of AP2, NAC, and GARP family TF binding correlations is enlarged on the right. Dashed boxes represent the homologous TFs in the heatmap, encompassing three sets of homoeologous genes. The homoeologous relationships of 189 TFs are listed in Supplementary Data 3. Source data are provided as a Source Data file.

repeats (~85%) in wheat genome, their expansion with built-in regulatory copies may quickly alter cognate TF binding patterns as reported in both plants and animals[21,24,27,37,38]. By overlapping with TEs, we detected 50–60% of the subgenome-specific TFBSs in TEs (Supplementary Fig. 6). The high sequence conservation and active epigenetic signature of TE-embedded TFBS indicated their functional relevance (Supplementary Fig. 7). 19,196 (11%) of TFBS with high chromatin accessibility reflected by seedlings DHS were embedded in TEs, representing highly active binding sites in vivo (Fig. 4a). TE-embedded TFBS without DHS may be active in response to specific developmental or external cues. An alternative but not mutually exclusive speculation is that TE-embedded TFBS evolved to promote TE propagation, which predisposed them to be co-opted for host gene regulation[21]. However, the contribution of TE-embedded TFBS to TE transcription under normal conditions may be limited, given the comparable transcription between TEs contributing to TFBS and TEs

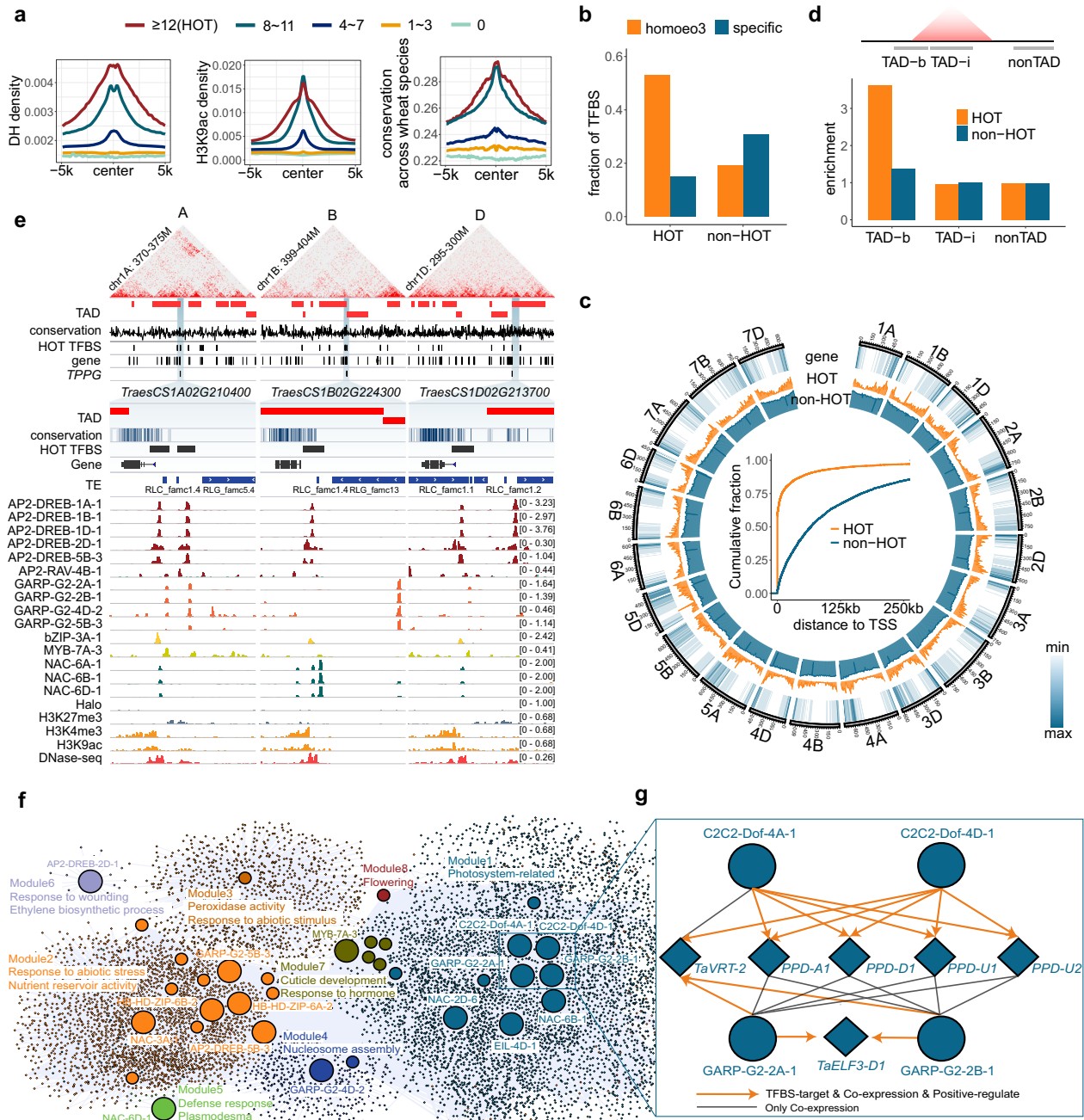

**Fig. 2 | Chromosomal features of the regulatory circuit. a** TFBSs were grouped according to the number of bound TFs. For each group, the average chromatin openness characterized by the DNase I hypersensitive site (DHS), the regulatory element activity characterized by the H3K9ac read density, and the sequence conservation across four wheat species with different ploidy levels are presented (see Methods). The average signal densities and conservation scores are indicated at a 100-bp resolution within a 10-kb window centered on the merged TFBS centers. Merged TFBSs with more than 12 binding sites were defined as high-occupancy target (HOT) regions. **b** Fractions of HOT and non-HOT regions overlapping subgenome-homoeologous regions and subgenome-specific regions, which were defined on the basis of the reciprocal alignment across subgenomes (see Methods). **c** Outer: Circos plot presenting the genomic distribution of gene density as well as HOT and non-HOT regions. Inner: cumulative fractions of the distance between a gene transcription start site and HOT regions or non-HOT regions. **d** Enrichment of HOT and non-HOT regions in the topologically

associating domain (TAD) boundaries (TAD-b), TAD internal regions (TAD-i), and non-TAD regions, which are illustrated above the bar plot, with the fraction of TAD-b, TAD-i, and non-TAD in the genome as the background, respectively. **e** Genomic tracks illustrating the targeting of the triad gene *TPPG* by a subset of TFs. The TAD regions, conservation scores across wheat species with different ploidy levels, TE locations, TF binding profiles, and the typical regulatory epigenetic marks, including H3K27me3, H3K4me3, and H3K9ac, are shown. The *TPPG* promoters are located in subgenome-homoeologous HOT regions and the TAD boundary. **f** Regulatory circuit integrating information from the TF-target pairs and co-expression modules. Node colors represent different co-expression modules. Circle and diamond nodes represent high- or median-confidence TFs and non-TFs, respectively. **g** Enlarged image of nodes. Lines with arrows represent targeted and positive co-expression relationships between TF and genes. Lines without arrows represent co-expression between TF and genes. Source data are provided as a Source Data file.

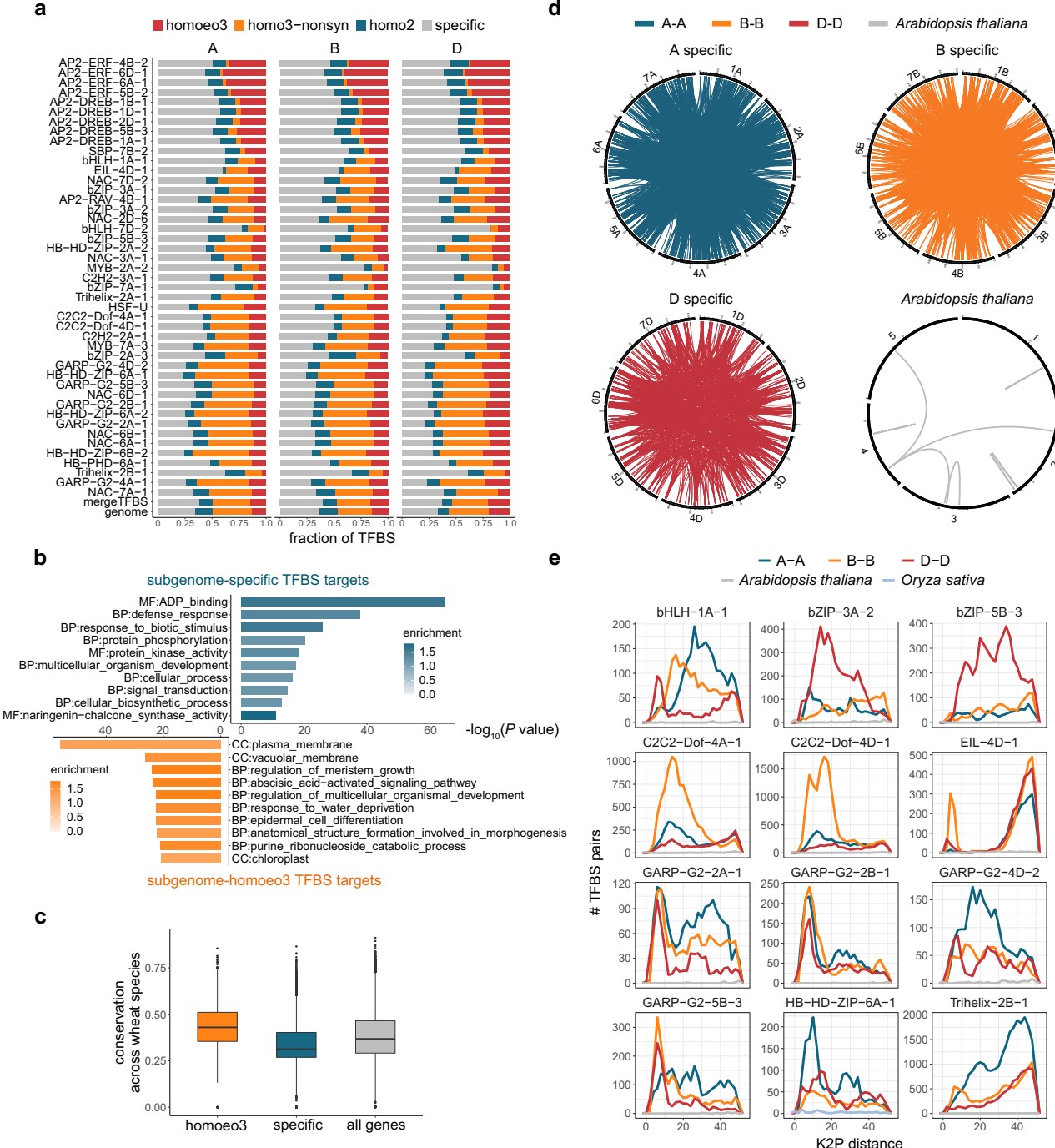

**Fig. 3 | Common wheat TFBSs underwent one or more round(s) of expansion.**
**a** Fraction of TFBSs in subgenome-homologous regions as determined by the reciprocal alignment across subgenomes. Homoeologou regions syntenic across three subgenomes were defined as homoeo3. **b** Most enriched GO terms for the subgenome-homoeologous and -specific TFBS-biased targeted genes ranked according to the enrichment $P$ value determined by two-tailed Fisher's exact test. The upper and lower bars represent high enrichment for subgenome-specific and -homoeologous TFBS targets, respectively. **c** Box plot showing the distribution of the conservation scores for subgenome-homoeologous ($n = 8140$) and -specific ($n = 14,540$) TFBS target genes and all genes (genes in the chrUn were removed, $n = 105,200$) across four wheat species with different ploidy levels. Horizontal lines in boxplots show median, hinges show interquartile range (IQR), whiskers show

$1.5 \times$ IQR, points beyond $1.5 \times$ IQR past hinge are shown. **d** Circos plots showing subgenome-specific bHLH-1A-1 TFBS pairs with high sequence similarity (BLASTN E-value <1e-30, identity >70%, and query coverage >70%) within each subgenome; the homolog bHLH28 in *Arabidopsis thaliana* is shown as a control. For each TF peak set, 400 peaks were randomly selected. **e** Sequence distance distribution between subgenome-specific TFBSs within each subgenome of CS and those of the corresponding TFs in *Arabidopsis thaliana* (gray line) and *Oryza sativa* (light blue line). The distance was calculated on the basis of the number of substitutions per 100 bases using the Kimura two-parameter (K2P) model[79]. According to this model, transitions (purine–purine and pyrimidine–pyrimidine) are more likely than transversions (purine–pyrimidine). Gaps are ignored. Source data are provided as a Source Data file.

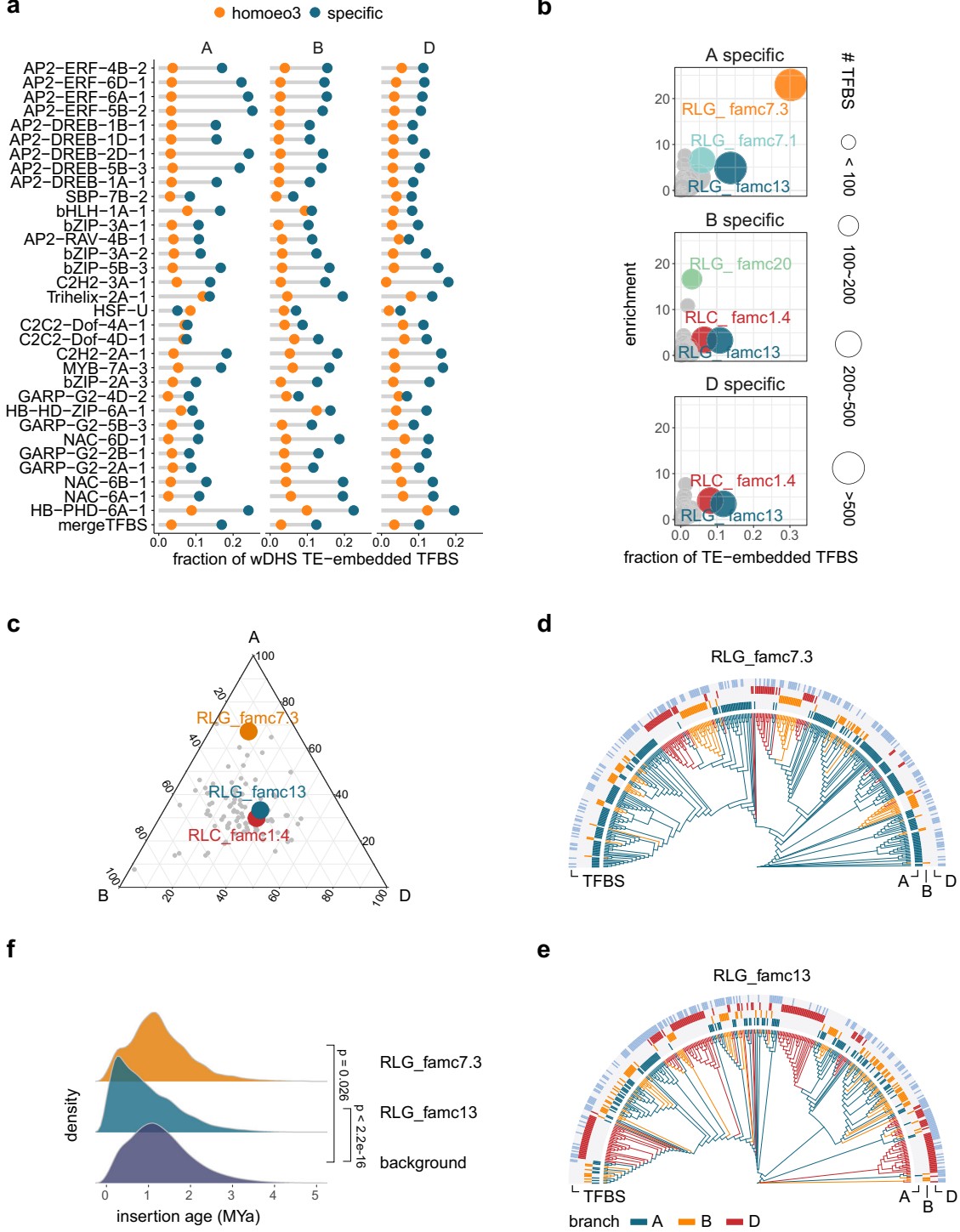

**Fig. 4 | Specific TE families contribute to subgenome-specific TFBS expansion.**
**a** Fraction of subgenome-homoeologous and -specific TFBSs embedded in TEs within open chromatin regions characterized by DHS. The TFs with more than 100 subgenome-homoeologous and -specific TFBSs within the DHS are presented.
**b** Enriched TE subfamilies harboring subgenome-specific TFBSs, with the fraction of each TE subfamily across the genome as the background. **c** Relative abundance of TE subfamilies across subgenomes. The most enriched TEs in Fig. 4b are highlighted. **d**, Dendrogram presenting the sequence similarity of the full-length

RLG_famc7.3 members. **e** Dendrogram presenting the sequence similarity of the full-length RLG_famc13 members. **f**. Age distribution of TEs enriched in the TFBSs determined according to the sequence similarity of the LTR at both ends. We corrected the K2P distance distribution by using the known evolutionary history as a reference: age = distance/(2× mutation rate), with a mutation rate of $1.3 \times 10^{-8}$ as previously described[10]. The two-tailed Wilcoxon signed-rank test was used to compare the LTR distance of different groups. Source data are provided as a Source Data file.

without TFBS (Supplementary Fig. 8). Regardless of the evolutionary forces driving the retention of TE-embedded TFBS, the large repertoire of subgenome-diversified TEs provides a rich source of TF occupancy for further evolutionary selection. Identifying these TE-derived TFBSs

provided a useful resource for further validating the regulatory effects of TEs in common wheat.

We next traced the expansion of TEs that contributed to TFBSs. TE families preferentially enriched among TFBSs present in only one

subgenome were detected. RLG_famc7.3 contributed a significant proportion of the subgenome A-specific TFBSs, whereas RLG_famc13 contributed to the divergent TFBSs in all three subgenomes (Fig. 4b, c). Similar results were obtained when using replicated data (Supplementary Fig. 9). These findings reflected the considerable plasticity of the regulatory framework shaped by TE-embedded TFBSs. The lineage-specific clustering of RLG_famc7.3 and RLG_famc13 was detected on the basis of the evolutionary tree, indicating specific expansions in different diploid progenitors (Fig. 4d, e). By using full-length LTRs to date the expansion events, we demonstrated that the recent expansions of both RLG families occurred after the divergence from the common diploid ancestor (~5 million years ago) (Fig. 4f).

### Effect of the unbalanced TF binding on target gene transcription

To characterize the regulatory consequences of subgenome-balanced and -unbalanced TF binding, we focused on gene-proximal TFBSs. Recent transcriptomic data indicated that at least 30% of subgenome-conserved triad genes (1:1:1 correspondence across three sub-genomes) exhibit unbalanced expression[18], which is likely coordinated by RE sequence contexts and epigenetic modifications[12]. To clarify this divergent regulation, we quantitatively partitioned TFBSs in triad promoters according to subgenome binding preference, and examined the expression of their target genes (Fig. 5a and Methods). The AP2 occupancy profile was stable across subgenomes, whereas the binding of GARP and NAC TFs were highly diverse (Fig. 5b, c). The subgenome-unbalanced binding of triads was consistent with the subgenome-unbalanced expression of these genes (Fig. 5d). Although not all of the differences in cis-regulatory sequences and transcription are associated with functional changes, possibly because of evolutionary drift[39], this diversity results in substantial raw genetic material for later uses, including the much later adaptations to environmental changes, as proposed by the 'radiation lag-time' hypothesis, which explains the observed delay between ancient polyploidization events and functional consequences[2,40].

### Balanced transcription mediated by parallel TFBS retention within asymmetrically decayed TEs

We further evaluated the impact of TEs on subgenome-balanced and -unbalanced TF bindings to triad promoters. Balanced and unbalanced TFBSs largely have similar fractions overlapping with TEs, with slightly higher ratios for balanced TFBSs (Fig. 6a, b). For triads with TE-embedded TFBSs in at least one promoter, more than 90% of both balanced and unbalanced TF binding was associated with TEs inserted in only one subgenome (Fig. 6c). It is unclear why the balanced TF occupancy is accompanied by the unbalanced contribution of TEs across triad promoters. We postulated that because of the long evolutionary history of TEs in wheat, there are many TE relics in the genome, which need to be considered when characterizing the dynamic effect of TE proliferations[10,27]. Accordingly, we searched for degenerated TE (dTE)-derived TFBS in triad promoters (Supplementary Fig. 10). For each triad with TE-derived TFBS in at least one promoter, the TE sequence was compared to promoter(s) of other gene(s) belonging to the same triad. The corresponding alignable promoter regions without canonical TE structures were defined as dTEs (Fig. 6d). When TEs and dTEs were considered together, the fraction of balanced triads with TE- and dTE-embedded TFBSs in all three triad members increased substantially (Fig. 6e, the orange box). For unbalanced triads, the contribution of dTEs to TFBSs was limited (Fig. 6e, the blue box). As a control, TEs contributed to TFBS from one triad promoter was compared to promoters of different triads without canonical TE structure. For each TF, 1000 permutation tests were conducted, and almost no alignable region was detected using the same cutoff as above (Supplementary Fig. 11). Furthermore, when comparing dTE-embedded TFBS with corresponding TE-embedded TFBS, ~90% TFBS sequences were highly consistent, suggesting a common origin

(Fig. 6f). Similar results were obtained when using DHS data reflecting in vivo binding activity, and were also supported by replicated data (Supplementary Fig. 12 and Supplementary Fig. 13). Thus, a considerable fraction of the balanced TFBSs derived from TEs may have degenerated, retaining only binding sites during evolution.

The biological importance of these dTE-derived TFBSs is supported by their high conservation across wheat species with different ploidy levels, i.e. diploid and tetraploid progenitors. However, the flanking TE sequences are highly diverse (Fig. 6g-j). Consistent with this result, the epigenetic profiles were indicative of active chromatin architecture at the dTE-derived TFBS but much less active in surrounding regions (Fig. 6j). This is an intriguing finding suggesting that a significant proportion of the TFBSs derived from anciently expanded TEs experienced parallel selection in each diploid lineage after divergence, whereas the flanking TE sequences were affected by relaxed selection or diversifying selection, resulting in unbalanced decay. Furthermore, by using DHS data to analyze the effect of TEs on RE activity and transcription, the specific evolutionary constraint on TE-derived REs and the apparent association between balanced RE activity and balanced expression were also detected (Fig. 6k, l). These results reflect the evolutionary effects of TE remnants on subgenome-convergent transcriptional regulation.

### Paleo-expansion of RLC_famc1.4 dominates TE-derived subgenome-convergent TFBS

Both RLC_famc1.4 and degenerated RLC_famc1.4 were highly enriched among the balanced TFBSs across triad promoters (Fig. 7a), accounting for 23% of the balanced TE-derived TFBSs. Notably, the ancient expansions of almost all TF families profiled herein were associated with amplification of RLC_famc1.4. A mixture of RLC_famc1.4 TEs from three subfamilies was detected in the phylogenetic tree (Fig. 7b), indicating most RLC_famc1.4 TEs may have been derived from the common ancestor of the diploid progenitors. To further trace the occurrence of RLC_famc1.4 expansion, we analyzed RLC_famc1.4 from Triticeae species, including *Secale cereale* (rye) and *Hordeum vulgare* (barley). The Kimura two-parameter (K2P) distances reflecting the genetic distance of RLC_famc1.4 between species were calculated. The distribution of the K2P distances between wheat subgenomes and between wheat and rye shared a peak centered on a similar K2P distance, suggesting that a paleo-expansion of RLC_famc1.4 occurred prior to the divergence of wheat and rye (Fig. 7c). In contrast, there were no common K2P distance peaks for RLG_famc7.3 and RLG_famc13, indicative of a lineage-specific expansion of these two subfamilies. The analysis of DHS data, which reflect chromatin openness and activity in vivo, also indicated that RLC_famc1.4 is the most enriched TE family for DHSs derived from both TE and dTE (Fig. 7d, e). Why this specific TE family dominates the TFBS exaptation in gene-proximal regions is an interesting issue. The possible mechanisms are discussed in the following section.

## Discussion

Cistrome maps for common wheat are a valuable resource for evaluating the integrated interactions of *cis*- and *trans*-factors that determine regulatory specificity. We revealed that diverse evolutionary forces acted on the paleo- and neo-TE-derived TFBSs, which mediate subgenome-divergent and -convergent TF binding, with distinct and synergistic regulatory consequences for the evolution of polyploids (Fig. 8).

Multiple TFBS expansion events were detected in wheat, but not in other model plants, including *A. thaliana* and *O. sativa* (Fig. 3d, e). This finding may be attributed to the active expansion of retroelements involving built-in TFBSs in wheat. The TE-embedded TFBS expansion events that occurred before and after the divergence of the diploid progenitors contributed to subgenome-common and -divergent TFBS expansion events, respectively, reflecting the importance of

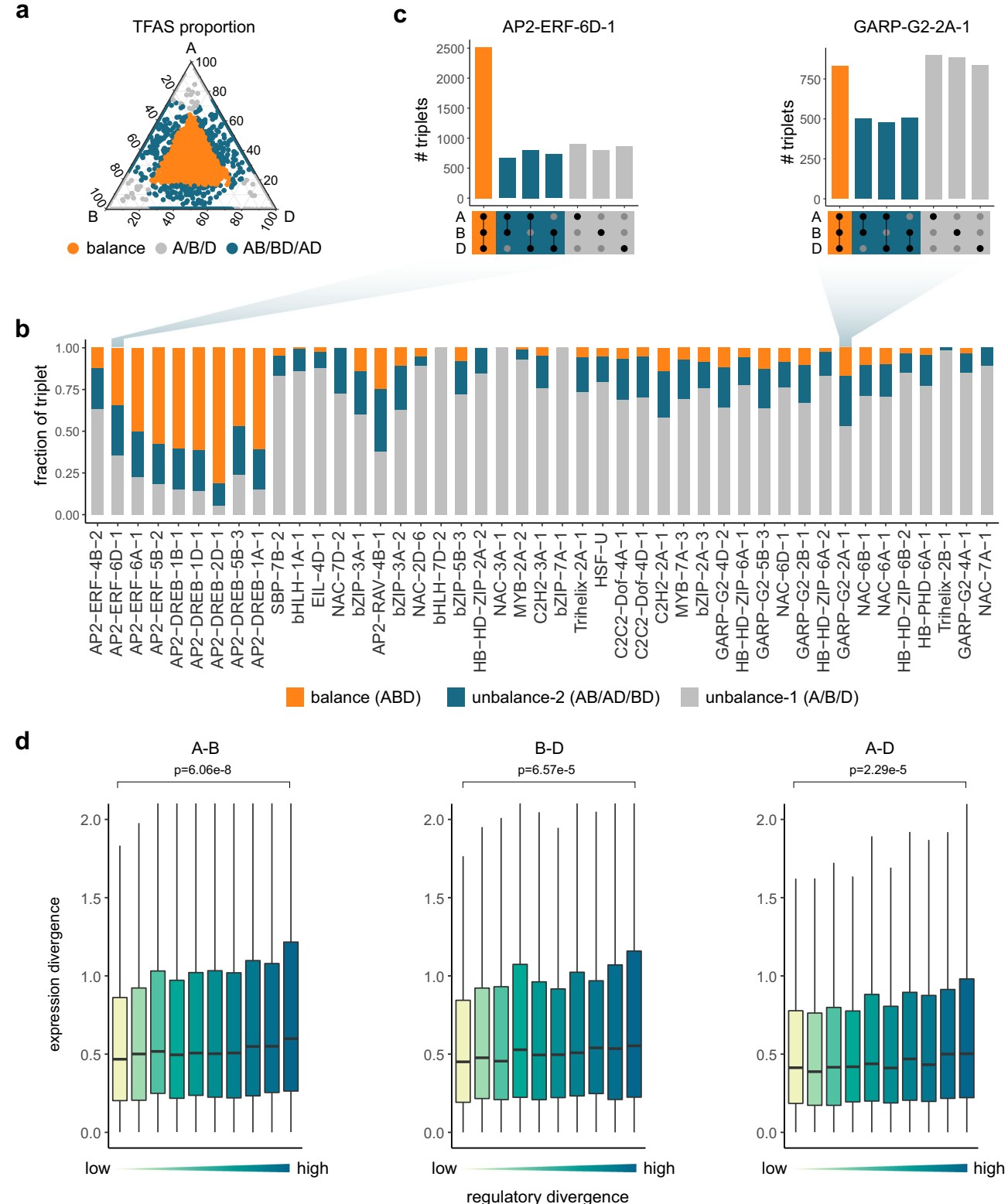

**Fig. 5 | Quantitative association between subgenome-biased TF binding and expression. a** Illustration of the quantitative divergence of TF binding in the promoters of triad genes. The relative binding of AP2-ERF-6D-1 is presented. Orange dots represent the balanced regulatory pattern. Blue and gray dots represent the unbalanced regulatory pattern. **b** Fraction of balanced and unbalanced TF binding in the promoters of triad genes (1:1:1 correspondence across subgenomes). **c** Bar plot presenting the results of the quantitative analysis of the balanced and unbalanced binding of AP2-ERF-6D-1 and NAC-6A-1. **d** Correlation of subgenome-biased binding and expression. Homoeologous gene pairs were grouped according to the regulatory divergence, which was calculated as the sum of the binding of all TFs | log$_2$(fold-change)| in the promoter regions. The expression divergence was calculated as the |log$_2$(fold-change)| of two orthologous genes. The two-tailed Wilcoxon signed-rank test was used to compare expression divergence between subgenomes A and B ($n = 8412$), B and D ($n = 6658$), A and D ($n = 8561$), respectively. Horizontal lines in boxplots show median, hinges show IQR, whiskers show 1.5 × IQR, points beyond 1.5 × IQR past hinge are shown. Source data are provided as a Source Data file.

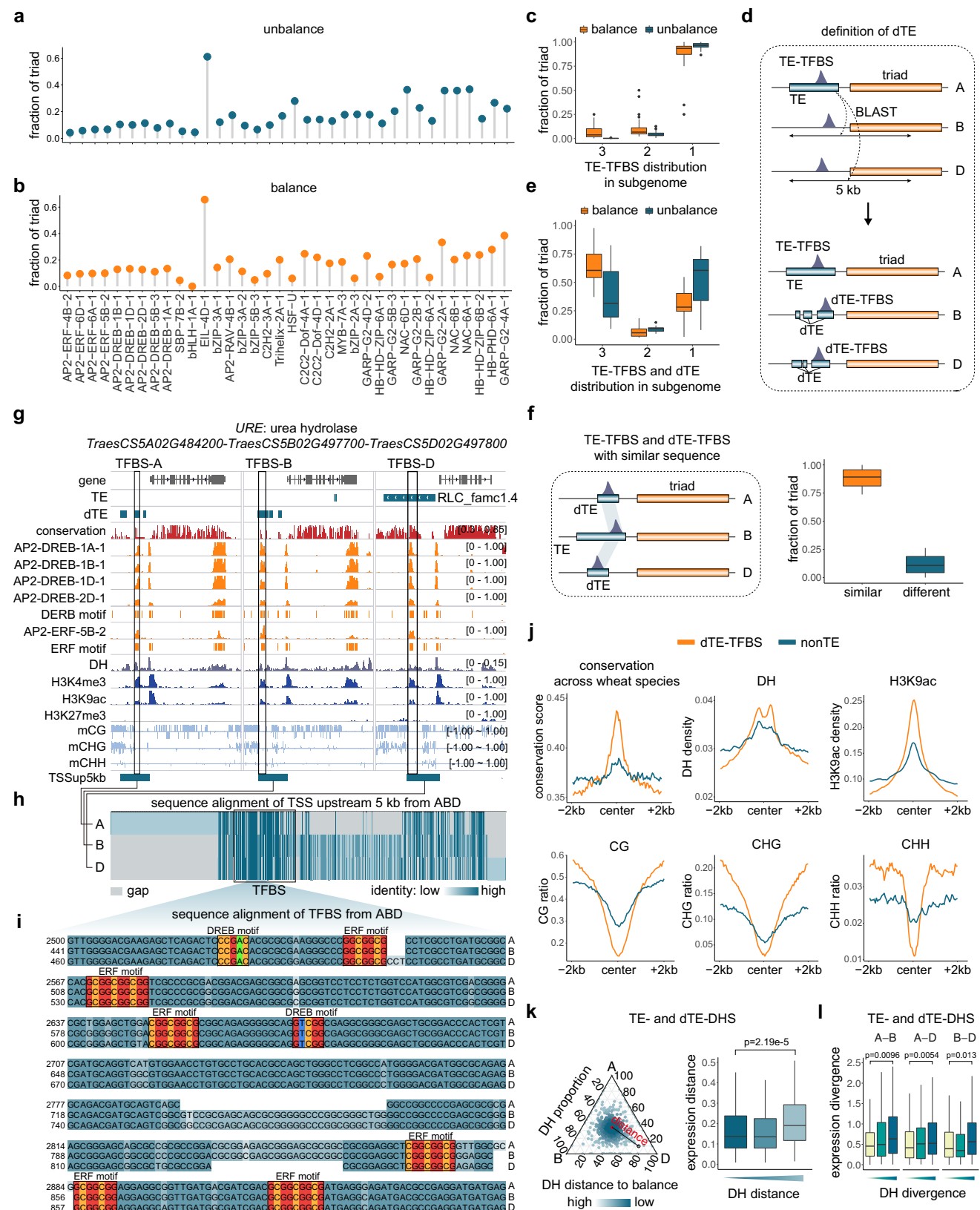

TE domestication for subgenome regulatory conservation and innovation. TEs contributed to TFBS were preferentially restricted to a limited number of TE families. RLC_famc1.4 expansions in common ancestor were associated with a significant fraction of the subgenome-homologous TFBS expansions (Fig. 7). Whereas RLG_famc7.3 spreadings unique to certain subgenome may have resulted in subgenome-

divergent regulation (Fig. 4). This is analogous to the human-specific dispersion of Alu elements, which participated in various human-specific regulatory events (e.g., conferring enhancer elements and modulating higher-order chromatin structures)[41]. Thus, TEs have significantly and continuously rewired wheat regulatory circuits. Following polyploidizations, *trans*-acting factors acquired additional suites of

**Fig. 6 | Subgenome-parallel retention of TE-derived TFBSs associated with balanced target gene transcription. a, b** Fraction of triads with unbalanced (**a**) and balanced (**b**) TF binding embedded in TEs. The TFs ($n = 36$) with >200 targeted triads are presented. **c** Fraction of triads with balanced and unbalanced TE-embedded TF binding in one, two, or three subgenomes. TFs ($n = 36$) targeting >200 triads are presented. Horizontal lines in boxplots show median, hinges show IQR, whiskers show $1.5 \times$ IQR, points beyond $1.5 \times$ IQR past hinge are shown. **d** Schematic for defining dTE and dTE-derived TFBS. **e** Fraction of triads with balanced and unbalanced TF binding embedded in TE and corresponding dTE present in one, two, or three subgenomes. TFs ($n = 25$) targeting >20 triads with TE-embedded TFBSs are presented. Boxplots definition is the same as Fig. 6c. **f** Left: schematic for identifying triads with similar TE and dTE-TFBS. Right: Fraction of triads with high sequence similarity of TFBS derived from TE and corresponding dTE ($n = 25$). Boxplots definition is the same as Fig. 6c. **g** Genome tracks illustrating dTEs contributing to the convergent TF binding to *URE* promoters. **h** Multiple

sequence alignment of the promoter region in Fig. 6g. **i** Alignment of the TFBS in Fig. 6h with AP2 motifs highlighted. **j** Conservation score and epigenetic profiles surrounding dTE-derived TFBSs and non-TE TFBSs. **k** Correlation between subgenome-biased chromatin openness of TE-embedded DHS and target gene expression. left: ternary plots showing the relative chromatin openness of **d** TE-embedded DHSs in triad promoters. The dot color represents the Euclidean distance of DHS density to the balanced point; right: triad genes were grouped according to the unbalanced level of promoter DHS, and the two-tailed Wilcoxon signed-rank test was used to compare triads ($n = 671$) expression distance. **l** Homoeolog pairs of triads were grouped according to the DH divergence, and the two-tailed Wilcoxon signed-rank test was used to compare expression divergence between subgenomes A and B ($n = 604$), B and D ($n = 489$), A and D ($n = 611$), respectively. Boxplots definition is the same as Fig. 6c. Source data are provided as a Source Data file.

cis-elements[42], which generated increasingly complex interactions potentially shaped by TEs. This considerable increase in the number of new interactions may have had an immediate or delayed effect on the adaptation of polyploid wheat[2,40].

Despite the extensive changes in intergenic regions across subgenomes by TE turnovers, the overall RE architecture was highly conserved, both in terms of the parallel evolution of TE-derived TFBSs and the extensive coordination of homoeologs (Fig. 6). Since the original report of TE functionalization by McClintock, there has been mounting evidence regarding the profound functional implications of TEs for the regulatory networks in animals[19–21] and plants[22–26]. However, TEs are subjected to rapid turnover, and the regulatory roles of TEs were mostly associated with creating new TFBSs. The TE relics and their evolutionary and functional importance are unclear, but they are crucial for deciphering the evolutionary effect of TEs on the genome-scale regulatory circuit. As a relatively young polyploid merged three highly plastic genomes shaped by abundant TEs of various ages, common wheat is ideally suited for subgenome comparisons aimed at clarifying the progressive and ongoing role of TE expansion and degeneration in regulatory evolution. A recent genome-wide characterization of common wheat TEs revealed that despite the intergenic turnover by TEs, unexpected preservation of the relative distance to genes was observed for specific TE families[10], implying certain TE families may have insertion preference relative to genes[43,44], some of which may have been commonly selected in different diploid progenitors. The present study demonstrated that the position-specific retention of TFBSs in specific TE families occurred in parallel across subgenomes. This apparent sequence conservation of TE-derived TFBS across subgenomes reflected their functional significance.

It is still a mystery why RLC_famc1.4 is the dominant TE family that contributed to TFBSs conserved across subgenomes. We studied this issue from perspectives of sequence and location. It was proposed in mammals that TEs with given built-in TF binding motifs tend to be favored by selection[37]. However, RLC_famc1.4 has no significant enrichment for the TF binding sequences compared to randomly selected TE regions (Supplementary Fig. 14). Mammalian studies demonstrated that TEs contributed to chromatin looping[45,46]. On the basis of the overlap with the local chromatin structure, we determined that RLC_famc1.4 sequences, particularly those overlapping TFBSs, were the most enriched sequences in TAD boundaries among the abundant TEs (Fig. 7f). This enrichment apparently applies only to subgenome-common TADs (Fig. 7g), indicating the parallel selection of TE-derived TFBSs may be associated with subgenome-convergent local chromatin structures.

## Method

### Plant materials and growth conditions

Common wheat [*Triticum aestivum* cultivar 'Chinese Spring' (CS)] seeds were surface-sterilized via a 10-min incubation in 30% $H_2O_2$

and then thoroughly washed five times with distilled water. The seeds were germinated in water for 3 days at 22 °C, after which the germinated seeds with a residual endosperm were transferred to soil. The seedlings were harvested after a 9-day incubation under long-day conditions. The above-ground parts of the harvested seedlings were frozen in liquid nitrogen for the DAP-seq assay.

### DAP-seq assay

Genomic DNA was extracted from wheat leaves using Plant DNAzol Reagent (Invitrogen) and then fragmented. The DNA ends were repaired using the End-It kit (Lucigen) and then an A-tail was added using the Klenow fragment (3′–5′ exo-; NEB). The truncated Illumina Y-adapter (Annealed by using adaptor strand A: 5′-ACACTCTTTCCC TACACGACGCTCTTCCGATCT-3′ and adaptor strand B: 5′-P-GATCG GAAGAGCACACGTCTGAACTCCAGTCAC-3′, where 'P' indicates a 5′ phosphate group) was ligated to the DNA using T4 DNA ligase (Promega). Full-length TF coding sequences were cloned into the pIX-Halo vector. For TFs with multiple isoforms, the longest coding sequence was selected. Halo-tagged TFs were expressed in vitro using the TNT SP6 Coupled Wheat Germ Extract System (Promega) and then immobilized using Magne HaloTag Beads (Promega) before they were incubated with the DNA library. The DNA binding to specific TFs was eluted for 10 min at 98 °C and then amplified by PCR using indexed Illumina primers and Phanta Max Super-Fidelity DNA Polymerase (Vazyme). To capture the background DNA, the Halo tag encoded in the empty pIX-Halo vector (i.e., without a TF coding sequence) was expressed and incubated with the DNA library. The amplified fragments were purified using VAHTS DNA Clean Beads (Vazyme) and then sequenced by Novogene (Beijing, China) using the Illumina NovaSeq 6000 system to produce 150-bp paired-end reads.

### Processing of DAP-seq, ChIP-seq, RNA-seq, and DHS data

We downloaded the histone ChIP-seq data for seven typical tissues and eight external stimuli, seedling RNA-seq data, and seedling DNase-seq data for CS from the NCBI GEO database (accession numbers GSE139019 and GSE121903)[11,12]. The *OsHOX24* ChIP-seq data for *Oryza sativa* were also obtained from the NCBI GEO database (accession number GSE144419)[47]. Additionally, the CS endosperm RNA-seq data were downloaded from the NCBI BioProject database (accession number PRJEB5135)[28]. Sequencing reads were cleaned using the fastp (version 0.20.0)[48] and Trim Galore (version 0.4.4) programs, which eliminated bases with low-quality scores (<25) and irregular GC contents as well as sequencing adapters and short reads. The remaining clean reads for the DAP-seq, ChIP-seq, and DHS analyses were mapped to the International Wheat Genome Sequencing Consortium (IWGSC) reference sequence (version 1.0) using the Burrows–Wheeler Aligner (version 0.7.17-r1188)[49]. The HISAT2 program (version 2.2.1)[50] was used for mapping the RNA-

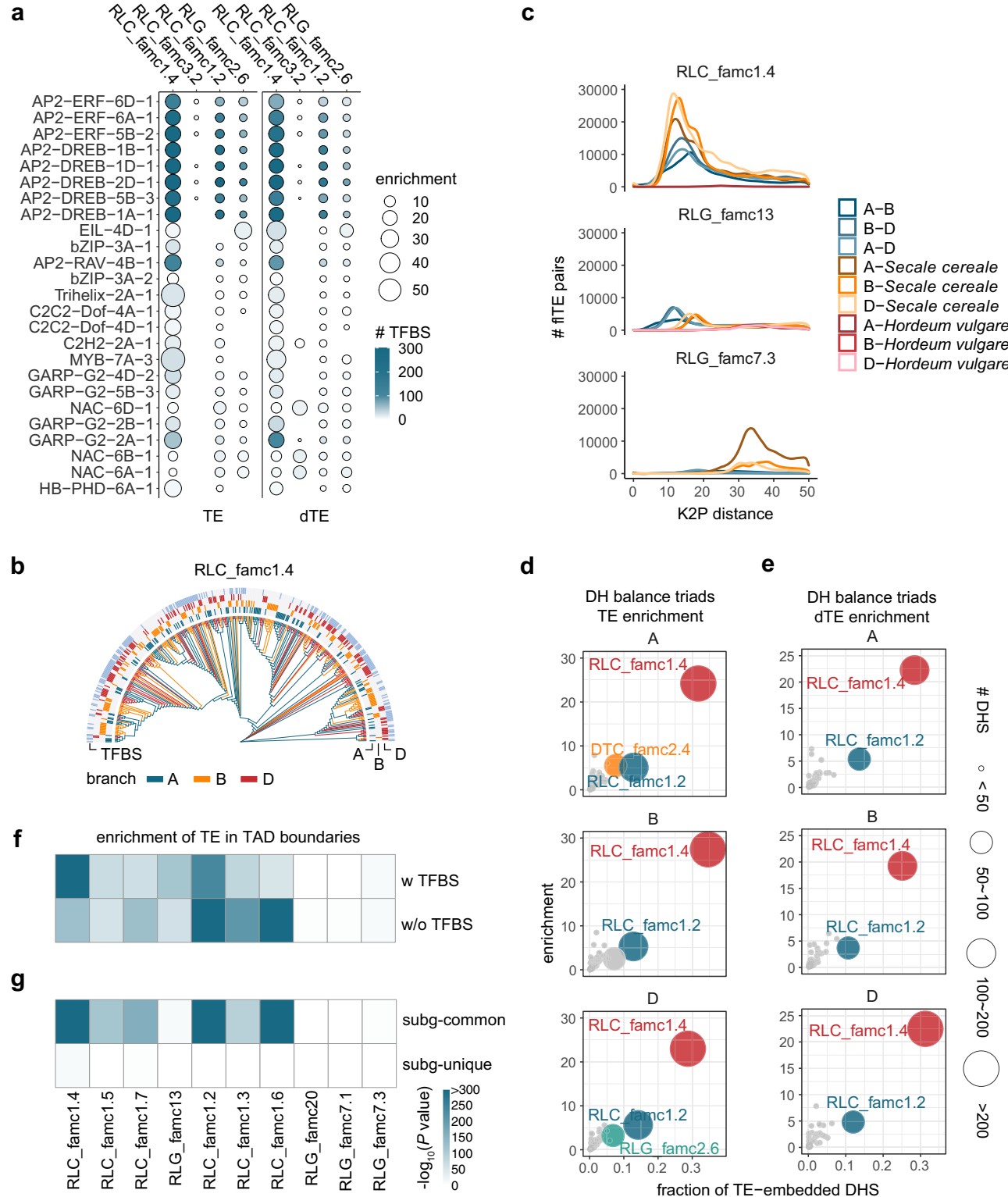

seq reads to the reference sequence. Reads with a mapping quality score <20 were removed. The remaining reads were mostly (>99.6%) mapped to only one position, and the multi-mapped reads were eliminated.

The MACS program (version 2.2.6)[51] was used to identify the read-enriched regions (peaks) on the basis of a threshold of $P < 1 \times 10^{-10}$. For the DAP-seq analysis, the peaks detected for the samples with the Halo tag alone were considered to represent non-specific binding (i.e., negative control). The TF peaks overlapping

the peaks detected for the Halo tag samples were excluded in the subsequent analysis. To quantify gene expression levels, the featureCount program of the Subread package (version 2.0.0)[52] was used to determine the RNA-seq read density for the genes. Integrative Genomics Viewer[53] was used to visualize the binding of TFs, histone markers, gene expression, and chromatin accessibility in the genome. The number of reads at each position was normalized against the total number of reads (i.e., reads per million mapped reads).

**Fig. 7 | Ancestral expansion of RLC_famc1.4 dominates TE-derived subgenome-convergent TFBSs. a** Enrichment of TE (left) and dTE (right) families that contributed to the balanced TF binding across triad promoters, with the fraction of TE in the genome as the background. **b** Dendrogram presenting the sequence similarity of the full-length RLC_famc1.4 members. **c** Sequence distance of full-length RLC_famc1.4, RLG_famc13, and RLG_famc7.3 members among three wheat subgenomes as well as between wheat and rye and wheat and barley. For each TE subfamily, 500 copies from subgenome A were randomly selected. The copy numbers for the B and D subgenomes as well as rye and barley were selected according to the total copy numbers in the B and D subgenomes as well as in rye and barley relative to the total copy number in subgenome A. For each TE copy, the sequences of the LTR at the 5' end and 3' ends were merged and aligned to other

merged LTRs. The copy numbers of the full-length RLC_famc1.4 and RLG_famc7.3 members in barley were <25. Thus, these sequences were not considered in this analysis. **d** Enrichment of TE families contributing to the balanced chromatin openness across triad promoters. The fraction of TE in the genome was used as background. **e** Enrichment of dTE families contributing to the balanced chromatin openness across triad promoters. The fraction of TE in the genome were used as background. **f** Enrichment of TE subfamilies in the TAD boundaries with and without TFBSs, with the fraction of TAD boundaries in the genome as the background. **g** Enrichment of TE subfamilies in the subgenome-common and -unique TAD boundaries, with the fraction of subgenome-syntenic and -unique TADs as the background, respectively. Source data are provided as a Source Data file.

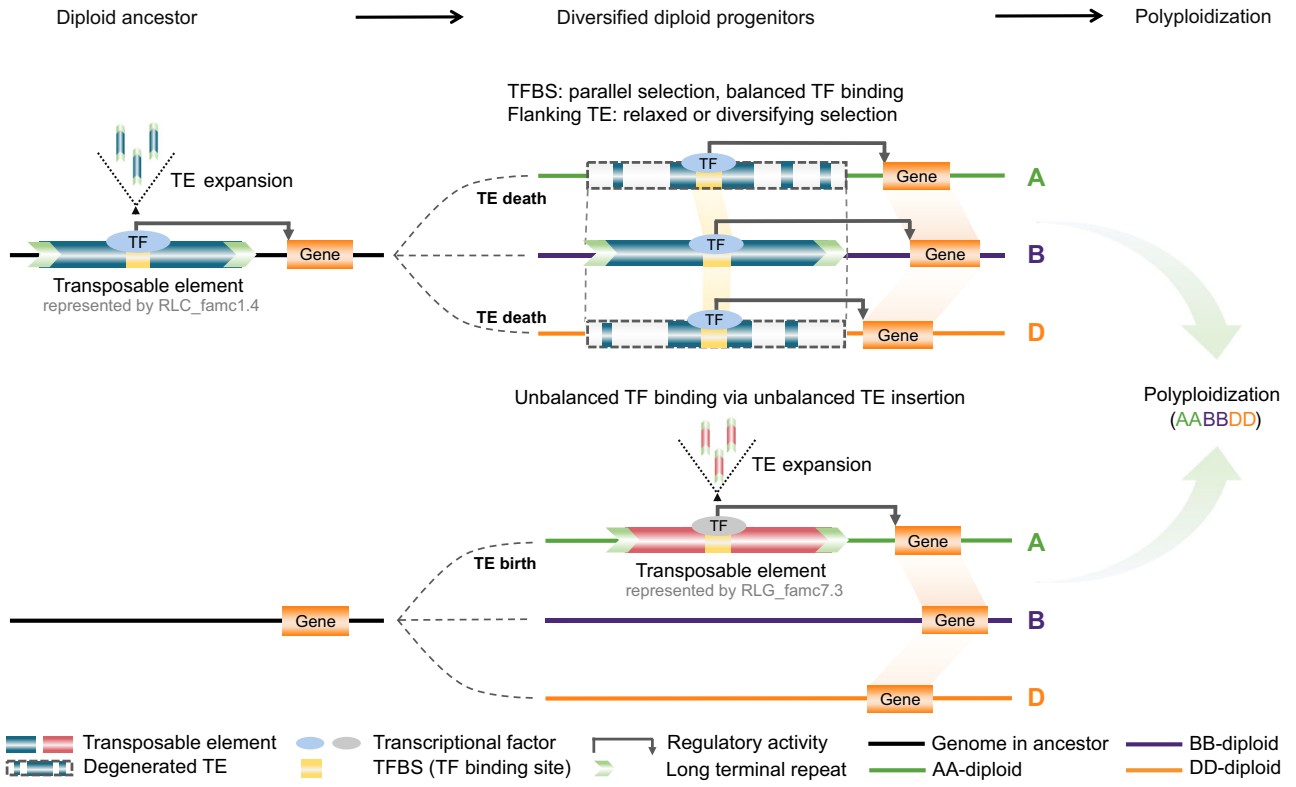

**Fig. 8 | Model illustrating the effects of TE-derived regulatory conservation and innovation on subgenome-convergent and -divergent TF binding and transcription.** Top: TE-embedded TFBSs expanded as TEs spread in the A-B-D common ancestor. In some cases, these TFBSs were maintained via parallel selection during the evolution of each diploid lineage after the divergence, whereas the flanking TE sequences degenerated to varying extents, resulting in conserved TFBSs and nonconserved flanking TE sequences. In hexaploid wheat, this balanced TF binding is associated with subgenome-balanced transcription. Bottom: TFBSs embedded in TEs differentially expanded in diploid lineages after the divergence from the common ancestor, which was responsible for the subgenome-divergent TF binding and variations in transcriptional regulation in polyploid wheat.

## Processing of Hi-C data

We downloaded the Hi-C data for CS[54] in the NCBI GEO database (accession number GSE133885). Reads were aligned to the IWGSC reference sequence (version 1.0) and filtered using HiC-Pro (version 2.11.1)[55]. The default parameter "-q 10" was used to retain uniquely mapped read pairs. We used "findTADsAndLoops.pl" implemented in the Homer software to detect TAD-like domains[56]. We used Juicer to generate KR-normalized contact matrices with bin sizes set to 25 kb and Juicerbox to visualize the TADs[57]. The TAD-like domain boundaries were identified as 20 kb regions centered at the boundary points.

## Detection and enrichment analysis of transcription factor-binding motifs

The peaks were sorted on the basis of the $q$-value and then the fold enrichment. The 600 bp sequence centered on the summit of the top 6000 peaks was used to detect de novo motifs using MEME-ChIP[58]

from the MEME software toolkit (version 5.1.1), whereas the enriched known motifs in the JASPAR database were detected using AME[59] from the MEME software toolkit. The de novo motifs were used to analyze the occurrence of individual motifs in the genome using the FIMO program[60] from the MEME software toolkit. Motif logos were drawn using the R package motifStack (version 1.34.0)[61] and universalmotif (version 1.4.0).

## Construction of a co-expression network

We downloaded 536 hexaploid wheat expression datasets from the Wheat Expression Browser (http://www.wheat-expression.com/)[18]. Genes with a TPM value <1 in at least 20 samples were removed and then 200 samples were randomly selected to generate a filtered expression matrix. Finally, 19,446 genes with high variance (top 25%) were retained. The WGCNA package (version 1.70.3)[62] was used to construct a co-expression network. An unsigned network was

constructed using the blockwiseModules function, with the following parameters: power = 6; maxModuleSize = 6000; TOMType = unsigned; minModuleSize = 30; reassignThreshold = 0; mergeCutHeight = 0.25; numericLabels = TRUE; and pamRespectsDendro = FALSE. If the co-expression partners of a gene could be defined by the above-mentioned criteria, they were assigned to the same module. Other-wise, the genes were classified in module 0. All edges were ranked according to the TOM value, and the top 80,000 edges were selected. The modules with HC and MC TFs (with 8,971 nodes) were visualized using Cytoscape (version 3.8.2)[63]. The GO terms curated by GOMAP[64] were used to detect the over-represented functional terms associated with the genes in each module.

## Calculation of the sequence conservation score

We completed a pair-wise comparison of the genome sequences using the NUCmer tool in the MUMmer package[65], with the parameter "--mum". For the comparison of diploid, tetraploid, and hexaploid wheat, the genome sequences of *Triticum urartu* (AA subgenome; IGDB version 1.0), *Aegilops tauschii* (DD subgenome; ASM34733 version 2), *Triticum turgidum* (AABB subgenome; WEWSeq version 1.0), and *T. aestivum* (AABBDD subgenome; IWGSC version 1.0) were used. The minimum sequence identity was set to 90 and each subgenome was treated as an individual genome. Next, ROAST[66] from multiz was used to integrate pair-wise sequence alignments into a multiple sequence alignment. The multiple sequence alignment and tree data were fitted using PhyloFit, after which the conservation score was calculated using phastCons from the PHAST package[67], with the parameters "--target-coverage 0.25; --expected-length 12".

## Detection of the subgenome-homologous and -specific regions

To determine the homologous regions across subgenomes, we used the subgenome alignment results generated by NUCmer. The reci-procal aligned regions that were longer than 400 bp were defined as homologous regions across three subgenomes (homo3) or two sub-genomes (homo2). Regions that were not aligned to another two subgenomes were defined as specific regions (specific). Subgenome-syntenic regions were detected using MCScanX (python version)[68], with homologous regions localized to syntenic regions defined as homoeo3, i.e., syntenic homo3 regions. Accordingly, 35%, 15%, 51%, and 16% of the genomic regions were defined as specific, homo2, homo3, and homoeo3, respectively.

## Sequence comparison of subgenome-specific TFBSs

The BLASTN algorithm was used to identify subgenome-specific TFBS pairs showing high sequence similarity within subgenomes, with the following parameters: E-value <1e-30, identity >70%, and query cov-erage >70%. The relationships among similar TFBS pairs (400 ran-domly selected TFBSs in each subgenome and *A. thaliana*) were visualized using Circos[69].

To analyze TFBS expansion, 500 randomly selected TFBS sequences in each subgenome for each TF were aligned using MAFFT (version 7.149b)[70]. The distance for each TFBS pair was calculated using 'Distmat' from EMBOSS (version 6.6.0.0)[71], which applied the widely used K2P model of nucleotide substitution for estimating genetic distance and phylogenetic relationships. The sequences of 500 randomly selected TFBSs for homologous TFs in *A. thaliana* were aligned and the distance was calculated in the same way.

## Enrichment of specific TE families that contributed to TF binding

We used CLARI-TE to annotate CS TEs. Additionally, ClariTeRep, which is a library containing the TEs and repeat sequences annotated in the TREP database and the annotated repeats on CS chromosome 3B[72], and RepeatMasker[73] were used to search the whole genome to detect candidate TEs. The results were prepared in an "embl" format to be used as the input file for CLARI-TE, which revealed the TE types, genomic positions, families, and subfamilies. The TE families were designated according to the rules of the ClariTeRep database. For example, RLG_famc7.1 and RLG_famc7.3 are subfamilies of RLG_famc7.

The TE subfamilies accounting for more than 0.1% of the genome length were selected. The enrichment scores (ES) for 98 TE subfamilies and 45 TFs were calculated using the following formula:

$$ES = \frac{\text{length of TF}(i)\text{peaks in TE subfamily}(j)/\text{length of all TF}(i)\text{peaks}}{\text{length of the TEs in subfamily}(j)/\text{length of all TEs in the genome}} \quad (1)$$

For the analysis of enriched TEs in the subgenome-homoeologous and -specific regions, the merged TFBSs for 45 TFs were used. To analyze the enrichment of dTEs, the non-degenerated TEs were used to calculate the length of the TEs in subfamily(j) and the length of all TEs in the genome.

## Evolutionary analysis of enriched TE subfamilies

LTRharvest[74] was used to identify the full-length LTRs of CS. Full-length TE sequences were aligned using MAFFT. FastTree (version 2.1.10) was used to construct the phylogenetic tree, which was visualized using the R package ggtree (version 2.4.1)[75]. The insertion time was determined on the basis of the divergence between the 5′ and 3′ LTRs and calcu-lated using distmat from EMBOSS.

## Definition of subgenome regulatory divergence

First, we determined the regulatory effect of each TF on each tar-get gene and then defined the subgenome regulatory divergence of each TF by comparing the regulatory effects on subgenome-homologous genes.

The regulatory effect was quantified according to the TF affinity score (TFAS), which was calculated using the following formula:

$$TFAS = \sum_{k=1}^{n} rpkm_k \times e^{-\frac{d_k}{2000}} \quad (2)$$

where $d$ is the distance from the peak summit to the gene transcription start site and $rpkm$ is the normalized read count (i.e., reads per kilobase per million mapped reads) in the peaks. The promoter was defined as the 5 kb region centered on the gene transcription start site. Addi-tionally, $n$ is the total number of peaks in the promoter. All $n$ peaks were considered to calculate the TFAS of a gene. The TFAS of the genes without a TFBS in the promoter was 0.

Orthofinder[76] was used to identify the orthologous genes between subgenomes. The orthogroups with only one copy in each subgenome (1:1:1) were defined as triads. The triads with a TFAS <0.25 for all three genes were filtered. We normalized the TFAS of the genes in each triad by calculating the proportion of the TFAS of one subgenome in the sum of three subgenomes. Subgenome-balanced and -unbalanced regulatory divergence patterns were represented by seven standard TFAS proportions. Specifically, the proportion [0.33, 0.33, 0.33] represented the balanced regulatory divergence pattern of the sub-genomes (ABD). The proportion [0.5, 0.5, 0], [0.5, 0, 0.5], [0, 0.5, 0.5] represented unbalanced regulatory pattern-2 (AB, AD, BD), whereas [1, 0, 0], [0, 1, 0], [0, 0, 1] represented unbalanced regulatory pattern-1 (A, B, D). The Euclidean distance from the normalized TFAS to the seven standard coordinates was calculated for each triad. The subgenome regulatory divergence pattern was assigned to the standard TFAS proportion pattern with the closest distance.

To compare the regulatory divergence and expression diver-gence, the regulatory divergence was quantified as the |log2(fold-change)| in the DAP-seq normalized read count for the promoters between subgenome 1:1 orthologous gene. Orthologous pairs with a TFAS greater than 0.25 for at least one gene were used. For each

orthologous pair, we summarized the regulatory divergence of all TFs that targeted the genes. The expression divergence was quantified as the |log$_2$(fold-change)| in CS seedlings between subgenome 1:1 orthologous gene.

The DNase-seq data were used to define the in vivo regulatory divergence. The chromatin openness score of each gene and the DH proportion of each triad with the TE-embedded DHS were calculated using the above-mentioned formula and method. We quantified the divergence of chromatin openness of each triad by calculating the Euclidean distance from the DH proportion to the standard balance point [0.33, 0.33, 0.33].

### Definition of dTEs

For triads with TE-embedded TFBS in at least one subgenomes, the BLASTN algorithm (version 2.9.0) was used to identify dTEs. Specifically, sequence of TE with TFBS in the promoters of one or two subgenomes were aligned with the promoters without canonical TE structures. Alignable regions overlapping with TEs were removed, and regions with alignment lengths > 50 bp were defined as dTE. For illustration in Fig. 6h–i, TE and dTE sequences in the *URE* promoter were aligned using MAFFT and visualized using Jalview (version 2.11.1.3)[77]. As a control, permutation tests were performed for each TF (Supplementary Fig. 11).

### Reporting summary

Further information on research design is available in the Nature Research Reporting Summary linked to this article.

## Data availability

The DAP-sequencing data generated in this study have been submitted to the NCBI Gene Expression Omnibus under accession number GSE192815. Tracks for all sequencing data can be visualized through our local genome browser [http://bioinfo.sibs.ac.cn/dap-seq_CS_jbrowse/]. Histone ChIP-seq data of seven typical tissues and eight external stimuli, RNA-seq and DNase-seq data of Chinese Spring(CS) seedling used in this study are under accession numbers GSE139019 and GSE121903 in NCBI GEO database[11,12]. Hi-C data of CS[54] used in this study is under accession number GSE133885 deposited in NCBI GEO database. The hexaploid wheat transcriptomic data of 536 samples used in this study were downloaded from Wheat Expression Browser [http://www.wheat-expression.com/][18]. The *OsHOX24* ChIP-seq data of *Oryza sativa* used in this study is under accession number GSE144419 deposited in NCBI GEO database[47]. The TFBS of *A. thaliana* used in this study were downloaded from Plant Cistrome Database [http://neomorph.salk.edu/dap_web/pages/index.php][78]. Source data are provided with this paper.

## Code availability

Scripts are available at Github [https://github.com/yuyun-zhang/hexa_dap].

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

## Acknowledgements

This study was supported by the Strategic Priority Research Program of the Chinese Academy of Sciences (XDB27010302)(Y.E.Z), National Science Fund for Excellent Young Scholars (32022012)(Y.J.Z), National Natural Science Foundation of China (31921005) (Y.B.X), and the National Key Research and Development Program (2021YFA1300404) (Z.B.L). We thank Dr. Jizeng Jia from Chinese Academy of Agricultural Sciences for insightful comments. We thank Huang Tao for his help in maintaining the high performance computing server.

## Author contributions

Y.J.Z., Y.B.X. and Z.B.L. conceived and designed the experiments. W.L.Z, Y.P.T., Z.J.L., L.H.Y., Y.P., H.S.D, Y.L.Z., W.T. and Y.E.Z performed the experiments. Y.Y.Z., J.Y.L., H.Y.W., Y.M., M.Y.W., Y.L.X., T.F.T., and Y.J.Z. analyzed the data. Y.J.Z. wrote the manuscript with input from all authors.

## Competing interests

The authors declare no competing interests.

## Additional information

[1]National Key Laboratory of Plant Molecular Genetics, CAS Center for Excellence in Molecular Plant Sciences, Shanghai Institute of Plant Physiology and Ecology, Shanghai Institutes for Biological Sciences, Chinese Academy of Sciences, 300 Fenglin Road, Shanghai 200032, China. [2]University of the Chinese Academy of Sciences, Beijing 100049, China. [3]State Key Laboratory of Genetic Engineering, Collaborative Innovation Center of Genetics and Development, Department of Biochemistry, Institute of Plant Biology, School of Life Sciences, Fudan University, Shanghai 200438, China. [4]The State Key Laboratory of Plant Cell and Chromosome Engineering, Institute of Genetics and Developmental Biology, the Innovative Academy of Seed Design, Chinese Academy of Sciences, Beijing 100101, China. [5]Shanghai Center for Plant Stress Biology, National Key Laboratory of Plant Molecular Genetics, Center of Excellence in Molecular Plant Sciences, Shanghai Institutes for Biological Sciences, Chinese Academy of Sciences, Shanghai 200032, China. [6]Henan University, School of Life Science, Kaifeng, Henan 457000, China. [7]State Key Laboratory for Crop Genetics and Germplasm Enhancement, Jiangsu Collaborative Innovation Center for Modern Crop Production, Nanjing Agricultural University, No.1 Weigang, Nanjing, Jiangsu 210095, China. [8]Institute of Advanced Biotechnology and School of Life Sciences, Southern University of Science and Technology, Shenzhen 518055, China. [9]Beijing Institute of Genomics, Chinese Academy of Sciences, and National Centre for Bioinformation, Beijing 100101, China. [10]Jiangsu Co-Innovation Center for Modern Production Technology of Grain Crops, Yangzhou University, Yangzhou 225009, China. [11]There authors contributed equally: Yuyun Zhang, Zijuan Li, Jinyi Liu, Yu'e Zhang, Luhuan Ye. ✉e-mail: zblang@cemps.ac.cn; ybxue@genetics.ac.cn; zhangyijing@fudan.edu.cn

