## [Peer Review File · Nature Communications]

Transposable elements orchestrate subgenome-convergent and -divergent transcription in common wheatReviewers' Comments:

Reviewer #1:

Remarks to the Author:

Review of Zhang et al., "Transposable elements orchestrate subgenome-convergent and -divergent transcription in common wheat."

Overview:

The authors have collected data on the DNA binding of >180 transcription factors across the wheat genome and assigned the corresponding transcription factor binding sites (TFBS) to the different subgenomes formed by the wheat polyploidies. They then assess which of the TFBS are shared and different between the subgenomes and argue that many of the "new" TFBS that are specific to one subgenome owe their origins to transposable element insertions.

Major comments:

The authors have collected a very powerful dataset for understanding a critical question, namely how gene expression evolves within a single genome after an allopolyploidy. Hence, I find the topic of this manuscript, the underlying data and the general conclusions relevant and important.

My first concern is that I think the authors need to be more careful with their evolutionary hypotheses. Two examples:

1) Allopolyploidies merge distinct genomes. I do not believe we can distinguish a TE bloom prior to allopolyploidy that produced a number of TFBS in a single ancestral genome that became one of the subgenomes from a similar bloom post polyploidy using the authors' data. But on the face of it, the former is much more likely, since then we need not explain why the bloom is specific to a subgenome. Perhaps this is in fact the authors' hypothesis for their data, but if so, they have explained it badly.

2) Not all transcription, nor all transcriptional differences, need be selectively meaningful: expression levels drift over time just as do sequences (1, 2). The fact that we observe a good deal of differences in subgenome transcription levels for homoeologous could be meaningless at the evolutionary level and without considering this possibility, the analyses here could be quite misleading.

Another of my concerns with the manuscript is that it very difficult to understand and, in particular, the authors write very loosely about their analyses in the Results section in a way that makes it very difficult to understand if those results are correct. Two examples:

Page 3: "80% of the HOT regions had highly conserved sequences among subgenomes."

I assume that the authors mean that, for a given HOT region in subgenome 1, they found the paralogous regions in subgenomes 2 and 3 using synteny and then aligned those regions to the original. But I am not at all certain that my interpretation is correct.

Page 4: "On average, 40% of the TFBS were localized in nonhomologous regions across subgenomes."

On its face, this sentence is nonsensical: if a given TF regulates 10 ancestral genes, each now in 3 copies, one in each subgenome, almost all of the TFBS will be nonhomologous, even if every gene and every site has been completely conserved after allopolyploidy (each gene has 2 paralogous TFBS from allopolyploidy and 9*3 nonhomologous ones). Again, I think what the authors are trying to say is that many TFBS have no homoeologous copy upstream of the homoeologous genes in the other two subgenomes. But I could be wrong.

There are also a good number of places where the writing is simply unclear:

Page 2: "Common wheat ... converged three subgenomes..."

Page 3: TAD is used without a definition

Page 4: A network of 34 TFs and 9000 genes is hardly "comprehensive" for the wheat genome

Finally, the authors have made some choices in their analysis that I don't understand:

1) Why were the DAP-seq data filtered to motifs (page 10)? If we know where the TF binds in the genome, not adding a potentially error-prone step of finding the motifs (which might be simply a human construct) could increase sensitivity.

2) Similarly, why was the coexpression network construction so focused on module identification? (page 10).

In short, I think there are some very interesting data here, but the manuscript is not ready for the community to take the most useful message from those data.

References:

1. Khaitovich P, Paabo S, & Weiss G (2005) Toward a neutral evolutionary model of gene expression. *Genetics* 170(2):929-939.

2. Yanai I, Graur D, & Ophir R (2004) Incongruent expression profiles between human and mouse orthologous genes suggest widespread neutral evolution of transcription control. *Omic: a journal of integrative biology* 8(1):15-24.

Reviewer #2:

Remarks to the Author:

The study investigates the role of TF binding and repetitive element evolution in regulation of duplicated homoeologous genes in the wheat genome. The reported work provides new insights into the mechanisms of homoeolog co-regulation in allopolyploid genome. Demonstration that TEs with TF-binding ability have potential to contribute to differential expression of duplicated genes in wheat is quite novel and critical for understanding the role of TE expansion on the evolution of wheat genome regulation.

I have some concerns about the presented work that need to be addressed.

L. 79-80. Quite strong statement and would require some references and more detailed explanation.

L. 86: Provide description of how many of TFs used in the study are duplicated homoeologous genes.

L.86: There is extensive variation in the number of isoforms generated by many genes in the wheat genome. It is not clear how the authors dealt with the TF isoforms which I have no doubt exist for the selected TFs. I suggest reporting the number of isoforms found for each TF, their relative abundance (which likely could be found in the public gene expression datasets) and which of the isoforms were used for this study.

L. 101: "...grouped together more than the TFBS for other TFs...". I am not sure what it means here. What criteria was used to compare grouping (besides visual assessment of the figure)? Based on figure, it seems that other TFBS also show grouping unique for each family.

L. 108: This sentence requires more detailed explanation. Why only H3K9ac was used? Was it used as a marker of active promoters? Not sure what the conservation across species means, how many species were used and how conservation was calculated. It should be clarified in the figure legend and y-axis label.

Fig. 3d. Not sure what distance on x-axis means – bp, kbp, Mbp?

L. 148: Provide similarity threshold and alignment lengths used to choose connections to create circos plots.

L. 161-162: I do not understand what the authors meant here. What is “evolutionary variance”? How was it calculated?

L. 191: Relatively small fraction of imbalanced triplets are associated with asymmetric TE insertion indicating that the mechanisms driving imbalanced expression can only partially be explained by differences in TF binding to TEs detected by the DAP-seq assay. I believe that trends detected in this study, while appear to be valid, should not be overinterpreted and presented like TE insertion is the main factor that define imbalanced expression.

L. 196: The conclusions in this paragraph are quite long reaching, undoubtedly very interesting, but I am not fully confident that presented data is sufficient to support this conclusion. Essentially, the authors claim that imbalanced expression of duplicated genes is caused by asymmetric insertion of TEs with TF-binding capacity into the regulatory regions. In the cases of balanced gene expression and the presence of asymmetric TE insertion, they claim that expression balance is achieved by both TEs and degenerated TEs with the TF-binding ability. The results of DAP-seq not always could be extrapolated to interactions happening on chromatin and ability of TFs to bind DNA sequences resembling TEs does not always mean that such interactions are happening in the cell. Building quite elaborate hypotheses seems to me require more direct evidence for the effects of TE domestication on chromatin and gene expression. I feel that in this study, the analyses of chromatin and epigenetic states are underutilized to explore deeper the mechanisms underlying imbalanced expression of triplets in wheat.

Please stick to correct usage of terminology - purifying selection. However, I do not think it is applicable in this case.

Minor comments:

1. I am not sure whether the references are allowed in the abstract.
2. Fig. 1C legend: Provide more detailed explanation how heatmap was generated.

Reviewer #3:

Remarks to the Author:

This is a very intriguing analysis of the evolution of (putative) TFBS in the large and complex wheat genome. Remarkably, the authors find evidence for massive expansion of these binding sites due to their frequent location within TEs, which have undergone rapid expansion in this genome. They go on to suggest that TEs, and specifically TFBSs within TEs, have contributed to the “plasticity” of wheat, presumably relative to other plants with fewer TEs. Although I find the quality of the analysis to be acceptable, I have serious issues with this interpretation. When dealing with TEs, these issues can arise from their sheer numbers. This can come from mapping of reads, which can be problematic when multiple TEs are nearly identical, too often-poor annotation of TEs. This impression is not helped by the fact that when discussing this annotation, the authors refer in their M&M to a second paper, which, in turn references a third paper. I don’t doubt that the short sequence motifs are frequently present in some TEs, and that some of them can even be captured by DAP-seq, although many would be left out depending on how repetitive sequences are masked. In looking at the M&M, it is also not clear when the DAP-seq data are being used and when enriched motifs present in the TEs are being used. Overall, I am concerned that the ways that the myriad difficulties associated with TE genomic analysis have been discussed.

The other concern I have is that the authors have consistently incorporated their conclusions into the text of the results. Although it is certainly possible, even likely, that TEs have contributed to gene regulation in wheat, I am deeply skeptical that they have done so, and I find it hard to believe that wheat is more plastic in its responses to the environment that it requires all of those additional TE-mediated regulations. Although the data are intriguing, I am not convinced that they have conclusively demonstrated the exaptation hypothesis. More broadly, I would suggest that the authors take the null hypothesis more seriously and treat their favored hypothesis with more skepticism.

Finally, the manuscript would benefit from extensive editing and the figure legends could provide some additional information, as indicated.

Specific comments:

Line 42. Although this is certainly an appealing hypothesis, the references do not in fact show that polyploid is the reason that wheat is a successful staple crop.

Line 68. Here and throughout, the manuscript could use editing for grammar.

Line 79. Wouldn't this just be a function of retention of regulatory elements that are required for tissue-specific expression in the homeologs? That is to say, why is it that the homeologs would need to be coordinated?

Line 89. From the homeologs?

Line 96. For readers not familiar with the assay, define success rate.

Line 100. I am curious as to what constitute positive and negative controls for these experiments, although I do see that the conservation between related TFs does make a good argument for conserved function. Very nice result!

Line 108. Epigenetic activity is a somewhat ambiguous term. Do you mean regions with chromatin marks consistent with transcriptional activity?

Line 112. Define TAD as "Topologically associating domains."

Figure 5. Define DHRs here or in associated text.

Line 118. Or that chromatin remodelers make these regions accessible? It would certainly be interesting to see in knockouts of any of these TFs affected TAD formation.

Figure 2. The legend here and elsewhere are somewhat sparse. I would suggest adding additional details. For instance, in figure 2d, what are TAD-b, TAD-I and not TAD with respect to the red lump on top of the line? Is this meant to be illustrative, or a presentation of actual data? And in figure 2b, enrichment relative to what? In 2e, although it is clear in context, you should say what is actually in the panel, including the fact that you are showing TF binding sites and chromatin marks, and that you are also showing TEs.

Line 135. This is very interesting, but it's hard to know exactly what it means unless we are told specifically how diverged the homeologs are in general.

Line 141. Again, interesting, but there is a lot of information here. Is there evidence that the components in these groups are in fact in conserved pathways? For instance, pathways regulated by

MADs box genes are relatively rapidly evolving, but those in other developmental processes may not be. And in fact, it looks as if developmental regulation is upregulated in both sets. This is an important observation, but it's not really fleshed out.

Line 144. "the subgenome-divergent regulatory circuit." should be "subgenome-divergent regulatory circuits."

Line 145. "We next asked the origin of subgenome divergent TF binding" Here and throughout, the manuscript would benefit from professional editing.

Line 147. Here, for instance, it's not clear what was done. The pairwise similarity of what with what? And what does "within each subgenome" mean? I think this means that each TFBS that lacked a homeolog in the other subgenomes was compared to all the others within each subgenome. I assume this means that the putative TFBS increased their copy number within the subgenomes. Does this mean the homologous TFBSs did not do so? And should I take it from the text that there are no additional similar TFBSs present on the other subgenomes? That is, not at the syntenic position or at any other position in the other subgenome?

Line 159. This is a huge and perhaps unjustified leap in logic. Clearly, the TEs have the motifs and clearly, when the TEs amplify, so those motifs will also amplify, and be similar to each other. That does not mean that these are actually functioning to regulate genes.

Line 162. The authors have provided no evidence for domestication and appear to be ignoring the null hypothesis.

Line 164. How was it determined that these are open chromatin "in vivo". Isn't this mapping of accessibility in chromatin preparations?

Line 170. This is only a hypothesis. No evidence has been provided that these actually function to regulate anything.

Line 187. To be clear, by triad promoters, you mean promoters of genes that are present in all three homeologs, and unbalanced means that although the promoters are there, the TFBS are not?

Line 212. This is very intriguing, and good evidence for functional relevance. This analysis would be enhanced if it were possible to know the age of these TEs. At least in some cases, could the LTRs be used to date these insertions; presumably they would be prior to the divergence of the subgenomes.

Line 216. It's hard to imagine how they could have no functional significance (at least historically) if they are conserved.

Figure 4I. This is a bit difficult to understand. The dTE TFBSs would be expected to be more conserved relative to each other than TFBSs that occur "naturally" because even when they are degenerate, the TE sequences flanking the transposon born TFBS would be expected to be more similar than the random sequences flanking the other TFBSs. It's also not clear whether or not the data set is biased, because of the search algorithm would pull out the subset of what may be a very large number of TEs in the same family that does not have conservation of this sequence, which is, after all, quite short.

Line 228. Because selection has favored this sequence for other reasons in this particular TE?

Line 264. Do we actually know that wheat is particularly "plastic" with respect to transcriptional regulation other than the changes that we would expect in any polyploid? What is the evidence for this?

Line 281. On first evolutionary principles, this argument makes very little sense. Any given insertion

carrying a random regulatory motif would almost certainly be selected against, regardless of its potential utility. If that is the case, then how would selection ever favor this kind of speculative "evolvability" function? Are the authors suggesting that selection somehow favored the evolution of TFBSs in the TEs in anticipation of their potential benefits?

We thank the reviewers for their supportive and constructive comments in helping us improve our manuscript. We now revised the vague descriptions and added necessary details. Please refer to the point-to-point response listed below:

REVIEWER COMMENTS

Reviewer #1 (Remarks to the Author):

Review of Zhang et al., “Transposable elements orchestrate subgenome-convergent and -divergent transcription in common wheat.”

Overview:

The authors have collected data on the DNA binding of >180 transcription factors across the wheat genome and assigned the corresponding transcription factor binding sites (TFBS) to the different subgenomes formed by the wheat polyploidies. They then assess which of the TFBS are shared and different between the subgenomes and argue that many of the “new” TFBS that are specific to one subgenome owe their origins to transposable element insertions.

Major comments:

The authors have collected a very powerful dataset for understanding a critical question, namely how gene expression evolves within a single genome after an allopolyploidy. Hence, I find the topic of this manuscript, the underlying data and the general conclusions relevant and important.

1. My first concern is that I think the authors need to be more careful with their evolutionary hypotheses. Two examples:

1) Allopolyploidies merge distinct genomes. I do not believe we can distinguish a TE bloom prior to allopolyploidy that produced a number of TFBS in a single ancestral genome that became one of the subgenomes from a similar bloom post polyploidy using the authors’ data. But on the face of it, the former is much more likely, since then we need not explain why the bloom is specific to a subgenome. Perhaps this is in fact the authors’ hypothesis for their data, but if so, they have explained it badly.

Response: We thank the reviewer for pointing this out. We realized that in the previous version, the “subgenome” and “diploid progenitor genome” are not clearly distinguished. Majority of subgenome divergence caused by TE expansion happened prior to the polyploidization events, mostly derived from diploid progenitors diverged from a common ancestor around 5 million years ago (DOI: 10.1126/science.1250092). We now revised the vague descriptions and the model in Figure 8.

a) Previous description “subgenome-specific” expansion of TEs is misleading, which is changed to specific expansion “in diploid progenitor” (lines 51-52, 69-70, 73, 192 and 312-313).

b) Model is now revised (Fig. 8). We clarified that the paleo TE expansion proposed in the manuscript indicates the TE proliferation occurred in the common ancestor of the three diploid progenitors (legend of Fig. 8).

figure 8

2) Not all transcription, nor all transcriptional differences, need be selectively meaningful: expression levels drift over time just as do sequences (1, 2). The fact that we observe a good deal of differences in subgenome transcription levels for homoeologous could be meaningless at the evolutionary level and without considering this possibility, the analyses here could be quite misleading.

Response: We thank the reviewer for the reminder and the literatures provided. We now added the discussion about transcriptional drift and the potential evolutionary impact of subgenome divergent TFBS. Please refer to lines 208-213.

2. Another of my concerns with the manuscript is that it very difficult to understand and, in particular, the authors write very loosely about their analyses in the Results section in a way that makes it very difficult to understand if those results are correct. Two examples:

1) Page 3: “80% of the HOT regions had highly conserved sequences among subgenomes.”

I assume that the authors mean that, for a given HOT region in subgenome 1, they found the paralogous regions in subgenomes 2 and 3 using synteny and then aligned those regions to the original. But I am not at all certain that my interpretation is correct.

Response: We reorganized the structure of the Results to make the logic clearer and added detailed descriptions.

1. Definition of homologous regions. In the previous version, the reciprocally aligned regions between two subgenomes were defined as homologous region. In the revised version, all three subgenomes were considered for sequence comparison,

and the proportions of homologous regions in syntenic and non-syntenic regions are calculated separately (lines 432-441 and Fig. 3).

2. HOT regions overlapped with homologous and syntenic regions across three subgenomes were defined as subgenome conserved HOT regions (lines 118-119).

2) Page 4: “On average, 40% of the TFBS were localized in nonhomologous regions across subgenomes.”

On its face, this sentence is nonsensical: if a given TF regulates 10 ancestral genes, each now in 3 copies, on in each subgenome, almost all of the TFBS will be nonhomologous, even if every gene and every site has been completely conserved after allopolyploidy (each gene has 2 paralogous TFBS from allopolyploidy and 9*3 nonhomologous ones). Again, I think what the authors are trying to say is that many TFBS have no homoeologous copy upstream of the homoeologous genes in the other two subgenomes. But I could be wrong.

Response: We thank the reviewer for the elaborative comment. In the manuscript, the homologous and nonhomologous regions in Figure 3 has no relationship with genes. As mentioned in the response to the above comment, homologous regions across subgenomes are conserved sequences across three subgenomes based on reciprocal alignment, with syntenic and non-syntenic regions calculated separately. We now described it in detail in both legend of Fig. 3a and the Methods section (lines 145-147, lines 432-441 and Fig. 3a).

3) There are also a good number of places where the writing is simply unclear:

Page 2: “Common wheat ... converged three subgenomes...”

Response: We thank the reviewer for pointing this out. We revised “*Common wheat converged three subgenomes...*” to “*Common wheat contains three sets of diploid genomes.*” (line 67). In addition, we updated the text where “subgenomes” needs to be replaced by “diploid genomes” (lines 51-52, 69-70, 73, 192 and 312-313).

Page 3: TAD is used without a definition

Response: The definition of TAD is added (line 122).

Page 4: A network of 34 TFs and 9000 genes is hardly “comprehensive” for the wheat genome

Response: we removed the ‘comprehensive’ in the revised manuscript.

3. Finally, the authors have made some choices in their analysis that I don’t understand:

1) Why were the DAP-seq data filtered to motifs (page 10)? If we know where the TF binds in the genome, not adding a potentially error-prone step of finding the motifs (which might be simply a human construct) could increase sensitivity.

Response: Our description about the filtering step may not clear enough which lead to some misunderstanding. We agree with the reviewer that a TFBS could be functional without the canonical binding motif. However, if the canonical motif is over-

represented in a given TF binding list, the reliability of this TF binding data could be relatively high. To draw robust conclusions from the data, for the analysis in the manuscript, we used a relatively strict criteria to select TFs with most of the binding sites contained the canonical motifs.

We acknowledged that those TFs with no significant fraction of bindings harbor typical motifs may also functional, as described in lines 100-101. Thus, for the purpose of a more general usage, we processed and deposited all TF binding data in public databases (GSE192815, reviewer token: appgkssuzvabdm), and provided all the genomic tracks in the customized genome browser (http://bioinfo.sibs.ac.cn/dap-seq_CS_jbrowse/).

2) Similarly, why was the coexpression network construction so focused on module identification? (page 10).

Response: We thank the reviewer for pointing this out. One main purpose of generating DAP-seq in the manuscript is to provide a directed regulatory network, and integrating co-expression network is to demonstrate the functional implication of these TF bindings. We now added the network construction using all TF targeting information (line 130-133, Fig. S3 and Table. S2) and revised the writing of this part.

In short, I think there are some very interesting data here, but the manuscript is not ready for the community to take the most useful message from those data.

Response: We re-organized the manuscript, made extensive editing, and added necessary analyses and detailed information in Results and legends. Please refer to the lines in blue.

References:

1. Khaitovich P, Paabo S, & Weiss G (2005) Toward a neutral evolutionary model of gene expression. *Genetics* 170(2):929-939.
2. Yanai I, Graur D, & Ophir R (2004) Incongruent expression profiles between human and mouse orthologous genes suggest widespread neutral evolution of transcription control. *Omics: a journal of integrative biology* 8(1):15-24.

Reviewer #2 (Remarks to the Author):

The study investigates the role of TF binding and repetitive element evolution in regulation of duplicated homoeologous genes in the wheat genome. The reported work provides new insights into the mechanisms of homoeolog co-regulation in allopolyploid genome. Demonstration that TEs with TF-binding ability have potential to contribute to differential expression of duplicated genes in wheat is quite novel and critical for understanding the role of TE expansion on the evolution of wheat genome regulation.

I have some concerns about the presented work that need to be addressed.

1) L. 79-80. Quite strong statement and would requires some references and more detailed explanation.

Response: We thank the reviewer for pointing this out. We now added references and changed “the primary mechanism” to “a primary mechanism”. Please refer to line 85.

2) L. 86: Provide description of how many of TFs used in the study are duplicated homoeologous genes.

Response: We summarized the homologous TFs in Table S3.

3) L.86: There is extensive variation in the number of isoforms generated by many genes in the wheat genome. It is not clear how the authors dealt with the TF isoforms which I have no doubt exist for the selected TFs. I suggest reporting the number of isoforms found for each TF, their relative abundance (which likely could be found in the public gene expression datasets) and which of the isoforms were used for this study.

Response: We thank the reviewer for pointing this out. We selected the isoform corresponding to the longest CDS (lines 346-347). We listed in Table S1 the number of isoforms found for each TF, their relative abundance and the isoforms used for this study.

4) L. 101: “...grouped together more than the TFBS for other TFs...”. I am not sure what it means here. What criteria was used to compare grouping (besides visual assessment of the figure)? Based on figure, it seems that other TFBS also show grouping unique for each family.

Response: We thank the reviewer for pointing this out. AP2 TFs are more compactly clustered together as reflected by higher darkness. However, the difference is not apparently large enough across TF families. Thus, we changed the description to “*Transcription factors from the same family generally had similar binding profiles*” (lines 108-109).

5) L. 108: This sentence requires more detailed explanation. Why only H3K9ac was used? Was it used as a marker of active promoters? Not sure what the conservation across species means, how many species were used and how conservation was calculated. It should be clarified in the figure legend and y-axis label.

Response: We thank the reviewer for pointing this out. H3K9ac is a typical active

marker in both promoter and enhancer in plants (doi.org/10.1186/s13059-019-1746-8, doi.org/10.1186/s13059-017-1273-4). We now added detailed explanation in lines 115-116 and the references.

The calculation of the conservation was described in Methods section “Calculation of the sequence conservation score”, which is now also described in the legend of Fig. 2a. Y-axis is now changed to ‘conservation across wheat species’.

6) Fig. 3d. Not sure what distance on x-axis means – bp, kbp, Mbp?

Response: We thank the reviewer for pointing this out. We revised the label of x-axis to “K2P distance”. The distance is calculated by the number of substitutions per 100 bases using the Kimura 2-parameter corrected model, based on the principle that transitions (purine–purine and pyrimidine–pyrimidine) are more likely than transversions (purine–pyrimidine). Gaps are ignored. (legend of Fig. 3e)

7) L. 148: Provide similarity threshold and alignment lengths used to choose connections to creat circus plots.

Response: We thank the reviewer for pointing this out. We now added the description of the threshold in line 443-446 and legend of Fig. 3d.

8) L. 161-162: I do not understand what the authors meant here. What is “evolutionary variance”? How was it calculated?

Response: We thank the reviewer for pointing this out. The previous statement is unclear, which is now changed to “*By overlapping with TEs, we detected 80%–90% of the subgenome-nonhomologous TFBSs in TEs (Fig. S7)*”. The detection of homologous and nonhomologous regions across subgenomes is described in Methods section. Please refer to lines 432-441 for details.

9) L. 191: Relatively small fraction of imbalanced triplets are associated with asymmetric TE insertion indicating that the mechanisms driving imbalanced expression can only partially be explained by differences in TF binding to TEs detected by the DAP-seq assay. I believe that trends detected in this study, while appear to be valid, should not be overinterpreted and presented like TE insertion is the main factor that define imbalanced expression.

Response: We agree with the reviewer’s comment. The impact of TEs on unbalanced TF binding in triad gene promoters is not large. We removed this part, merged and simplified the results to make the logic clearer. Please refer to lines 218-224.

10) L. 196: The conclusions in this paragraph are quite long reaching, undoubtedly very interesting, but I am not fully confident that presented data is sufficient to support this conclusion. Essentially, the authors claim that imbalanced expression of duplicated genes is caused by asymmetric insertion of TEs with TF-binding capacity into the regulatory regions. In the cases of balanced gene expression and the presence of asymmetric TE insertion, they claim that expression balance is achieved by both TEs

and degenerated TEs with the TF-binding ability. The results of DAP-seq not always could be extrapolated to interactions happening on chromatin and ability of TFs to bind DNA sequences resembling TEs does not always mean that such interactions are happening in the cell. Building quite elaborate hypotheses seems to me require more direct evidence for the effects of TE domestication on chromatin and gene expression. I feel that in this study, the analyses of chromatin and epigenetic states are underutilized to explore deeper the mechanisms underlying imbalanced expression of triplets in wheat.

Response: We thank the reviewer for this suggestion. We used DNase I hypersensitive sites (DHS), representing active regulatory regions *in vivo*, to evaluate the impact of TEs on chromatin openness and association with target gene expression. Similar results were obtained in terms of the impact of ancient TE expansion on balanced binding and gene transcription (lines 232-234, 244-247, Fig. 6j and Supplementary Fig. S9-10).

Evaluation of the impact of TE- and dTE-derived REs on subgenome balanced regulation and transcription using DHS data (Fig. 6j and Supplementary Fig. S9-10)

12) Please stick to correct usage of terminology - purifying selection. However, I do not think it is applicable in this case.

Response: We thank the reviewer for pointing this out. The usage of “purifying selection” without evolutionary test is inappropriate. We changed the “*parallel purifying selection*” to “*parallel selection*” (line 242).

Minor comments:

1. I am not sure whether the references are allowed in the abstract.

Response: We thank the reviewer for pointing this out. The references in the abstract are removed.

2. Fig. 1C legend: Provide more detailed explanation how heatmap was generated.

Response: We added more details to the legend of Fig. 1c. “*The genome is divided into consecutive 2 kb bins. For each TF, bins are binarized, indicating the presence or absence of TFBSs, which are used to calculate the Pearson correlation coefficient.*”

Reviewer #3 (Remarks to the Author):

This is a very intriguing analysis of the evolution of (putative) TFBS in the large and complex wheat genome. Remarkably, the authors find evidence for massive expansion of these binding sites due to their frequent location within TEs, which have undergone rapid expansion in this genome. They go on to suggest that TEs, and specifically TFBSs within TEs, have contributed to the “plasticity” of wheat, presumably relative to other plants with fewer TEs. Although I find the quality of the analysis to be acceptable, I have serious issues with this interpretation. When dealing with TEs, these issues can arise from their sheer numbers. This can come from **mapping of reads**, which can be problematic when multiple TEs are nearly identical, too often-poor annotation of TEs. This impression is not helped by the fact that **when discussing this annotation**, the authors refer in their M&M to a second paper, which, in turn references a third paper. I don't doubt that the short sequence motifs are frequently present in some TEs, and that some of them can even be captured by DAP-seq, although many would be left out depending on how repetitive sequences are masked. In looking at the M&M, it is also not clear **when the DAP-seq data are being used and when enriched motifs present in the TEs are being used**. Overall, I am concerned that the ways that the myriad difficulties associated with TE genomic analysis have been **discussed**.

Response: We thank the reviewer for pointing this out.

1. A stringent cutoff of read mapping was applied. Reads with a mapping quality score less than 20 were removed. The remained reads were mostly (> 99.6%) mapped to only one position, and the multi-mapped reads were eliminated. (lines 371-373 in Methods).
2. We added detailed description about how TE annotation was performed (lines 455-462 in Methods).
3. For all the motif analysis, only motifs in TFBS detected by DAP-seq were collected. We added detailed description about motif detection (line 97, and legend Fig. 1b).

The other concern I have is that the authors have consistently incorporated their conclusions into the text of the results. Although it is certainly possible, even likely, that **TEs have contributed to gene regulation in wheat**, I am deeply skeptical that they have done so, and I find it hard to believe that wheat is more plastic in its responses to the environment that it requires all of those addition TE-mediated regulations. Although the data are intriguing, I am not convinced that they have conclusively demonstrated the exaptation hypothesis. More broadly, I would suggest that the authors take the null hypothesis more seriously and treat their favored hypothesis with more skepticism.

Response: We agree with the reviewer's comments. We made the following revisions.

1. We deleted all conclusions about the subgenome plasticity caused by TE-derived TFBS, and the conclusions about the regulatory role of TE were limited to the contribution of subgenomic polymorphisms. For example, in lines 189-190, “*These findings reflected the considerable diversity of the regulatory framework shaped by TE-embedded TFBS.*”

2. We discussed the relationship between subgenome diversity and transcriptional drift, as well as the impact on regulatory evolution (lines 208-213).

Finally, the manuscript would benefit from extensive editing and the figure legends could provide some additional information, as indicated.

Response: We re-organized the manuscript, made extensive editing, and added necessary analyses and detailed information in Results and legends. Please refer to the lines in blue.

Specific comments:

Line 42. Although this is certainly an appealing hypothesis, the references do not in fact show that polyploid is the reason that wheat is a successful staple crop.

Response: We thank the reviewer for pointing this out. These three references were incorrectly retained. We have now added relevant references to the Introduction section (line 69), and removed these references from the Abstract.

Line 68. Here and throughout, the manuscript could use editing for grammar.

Response: We carefully checked the manuscript and corrected the grammar errors.

Line 68 in previous version is changed to “*leading to the development of common wheat as a staple crop cultivated worldwide*” (lines 73-74).

Line 79. Wouldn't this just be a function of retention of regulatory elements that are required for tissue-specific expression in the homeologs? That is to say, why is it that the homeologs would need to be coordinated?

Response: In the revised manuscript, this statement is changed to “*despite the highly diverse regulatory regions, earlier research revealed the extensive balanced expression of homeologs throughout development, raising an additional question regarding how this evolutionary constraint on transcriptional regulation was achieved*” (lines 81-84).

Line 89. From the homeologs?

Response: We thank the reviewer for pointing this out, and changed “homologs” to “homeologs” (line 95).

Line 96. For readers not familiar with the assay, define success rate.

Response: We thank the reviewer for pointing this out. The success rate is “*represented by the fraction of HC TFs for each TF family*” (lines 103-104).

Line 100. I am curious as to what constitute positive and negative controls for these experiments, although I do see that the conservation between related TFs does make a good argument for conserved function. Very nice result!

Response: We thank the reviewer for the positive comment. In DAP experiment, the TF is expressed in fusion with the Halo-tag, and the peaks detected in samples with only Halo-tag introduced were considered as non-specific bindings (negative control). TF peaks overlapping with peaks detected from Halo samples were removed for

subsequent analysis (Methods, lines 375-378). Since DAP-seq reflects the direct DNA binding ability of TFs, the enrichment of cognate motifs reflects the fidelity of the experiment, which is used as the criteria determining the high and low confidence of the experiment.

Line 108. Epigenetic activity is a somewhat ambiguous term. Do you mean regions with chromatin marks consistent with transcriptional activity?

Response: We thank the reviewer for pointing this out. We changed “epigenetic activity” to “*H3K9ac activity typical of active promoters and enhancers in wheat*” (line 116), which is a typical marker for active REs including both promoters and enhancers in plants (wheat: doi.org/10.1186/s13059-019-1746-8; maize: doi.org/10.1186/s13059-017-1273-4).

Line 112. Define TAD as “Topologically associating domains.”

Response: We thank the reviewer for pointing this out. The TAD is now defined (line 122).

Figure 2. Define DHRs here or in associated text.

Response: We now added the description of DHS (DNaseI hypersensitive site) in line 115 and legend of Figure 2a.

Line 118. Or that chromatin remodelers make these regions accessible? It would certainly be interesting to see in knockouts of any of these TFs affected TAD formation.

Response: It is true that the cause and effect cannot be distinguished merely based on association. We changed the conclusion to “*implies that a high TF occupancy may be associated with TAD formation. Alternatively, the chromatin architecture in TAD boundaries may help facilitate TF occupation*” (lines 127-129).

Figure 2. The legend here and elsewhere are somewhat sparse. I would suggest adding additional details. For instance, in figure 2d, what are TAD-b, TAD-i and not TAD with respect to the red lump on top of the line? Is this meant to be illustrative, or a presentation of actual data? And in figure 2d, enrichment relative to what? In 2e, although it is clear in context, you should say what is actually in the panel, including the fact that you are showing TF binding sites and chromatin marks, and that you are also showing TEs.

Response: We now added more detailed information in the legends of Fig. 2d and 2e.

Figure 2d, Enrichment of HOT and non-HOT regions in the topologically associating domain (TAD) boundaries (TAD-b), TAD internal regions (TAD-i), and non-TAD regions, which are illustrated above the bar plot, with the fraction of TAD-b, TAD-i, and non-TAD in the genome as the background, respectively.

Figure 2e, Genomic tracks illustrating the targeting of the 1:1:1 homeologous gene TPPG by a subset of TFs. The TAD regions, conservation scores across wheat species

with different ploidy levels, TE locations, TF binding profiles, and the typical regulatory epigenetic marks, including H3K27me3, H3K4me3, and H3K9ac, are shown. The TPPG promoters are located in subgenome-homologous HOT regions and the TAD boundary.

Line 135. This is very interesting, but it's hard to know exactly what it means unless we are told specifically how diverged the homeologs are in general.

Response: We now added details about the definition and fraction of homologous regions across subgenomes in line 432-441 (Methods).

Line 141. Again, interesting, but there is a lot of information here. Is there evidence that the components in these groups are in fact in conserved pathways? For instance, pathways regulated by MADs box genes are relatively rapidly evolving, but those in other developmental processes may not be. And in fact, it looks as if developmental regulation is upregulated in both sets. This is an important observation, but it's not really fleshed out.

Response: We thank the reviewer for pointing this out. We updated the definition of homologous and nonhomologous regions (lines 432-441), and re-analyzed the enriched GO terms of the targets (Fig. 3b). Similar results were obtained. We now revised the writing according to the suggestion to “*The most enriched GO term among the genes with homologous (homo3-syn) TFBSs was membrane architecture (Fig. 3b), whereas genes with subgenome-divergent TFBSs were mostly related to defense, with sequences that varied among wheat species (Fig. 3c)*”. We used the conservation of target genes across wheat species to reflect the conservation of the pathway (Fig.3c). (lines 152-155).

Line 144. “the subgenome-divergent regulatory circuit.” should be “subgenome-divergent regulatory circuits.”

Response: We thank the reviewer for pointing this out. It is now corrected (line 156).

Line 145. “We next asked the origin of subgenome divergent TF binding” Here and throughout, the manuscript would benefit from professional editing.

Response: We reorganized the structure of the Results to make the logic clearer and added detailed descriptions. Please refer to the sentences in blue.

Line 147. Here, for instance, it's not clear what was done. The pairwise similarity of what with what? And what does "within each subgenome" mean? I think this means that each TFBS that lacked a homeolog in the other subgenomes was compared to all the others within each subgenome. I assume this means that the putative TFBS increased their copy number within the subgenomes. Does this mean the homologous TFBSs did not do so? And should I take it from the text that there are no additional similar TFBSs present on the other subgenomes? That is, not at the syntenic position or at any other position in the other subgenome?

Response: We thank the reviewer for the elaborative comment. We added detailed description about the definition of homologous regions. In the previous version, the reciprocally aligned regions between two subgenomes were defined as homologous region. In the revised version, all three subgenomes were considered for sequence comparison, and the proportions of homologous regions in syntenic and non-syntenic regions are calculated separately (lines 145-147, lines 432-441 and Figure 3). Thus, the nonhomologous region in the revised Fig. 3c are subgenome specific and not at the syntenic position or at any other position in the other subgenomes (line 158).

Line 159. This is a huge and perhaps unjustified leap in logic. Clearly, the TEs have the motifs and clearly, when the TEs amplify, so those motifs will also amplify, and be similar to each other. That does not mean that these are actually functioning to regulate genes.

Response: We agree that this conclusion is too strong to be justified. This sentence was to introduce the published findings indicating TEs with built-in motifs may lead to genome-wide transcriptional innovation. We now revised the previous sentence "*expansion of which with built-in regulatory copies may quickly **rewire transcriptional patterns** and lead to novel functions and increasing regulatory complexity*" to "*expansion of which with built-in regulatory copies may quickly **alter cognate TF binding patterns***" (lines 176-177).

In addition, the published studies are now moved to the last part of this paragraph. "*Recent reports implied that TE expansion and exaptation may lead to novel transcriptional regulation and increased regulatory complexity. Identifying these TE-derived TFBSs may be useful for further validating the regulatory effects of TEs on the polyploidization of wheat*" (lines 181-184).

Line 162. The authors have provided no evidence for domestication and appear to be ignoring the null hypothesis.

Response: This sentence is now revised to be clearer and more direct "*By overlapping with TEs, we detected 80%–90% of the subgenome-nonhomologous TFBSs in TEs*" (lines 177-178).

Line 164. How was it determined that these are open chromatin "in vivo". Isn't this

mapping of accessibility in chromatin preparations?

Response: The open chromatin is determined by DNaseI hyper sensitive site (DHS), which reflects the chromatin openness *in vivo*. The data was published previously (doi: doi.org/10.1186/s13059-019-1746-8). We now added the details in lines 115-116 and legend of Fig. 2a.

Line 170. This is only a hypothesis. No evidence has been provided that these actually function to regulate anything.

Response: We agree with the reviewer's comment, and changed the statement of "*The enrichment of different TE families in subgenome-specific TFBS indicates the high plasticity of the regulatory framework was shaped by TE-embedded TFBS.*" to "*These findings reflected the considerable diversity of the regulatory framework shaped by TE-embedded TFBSs.*" (lines 189-190).

Line 187. To be clear, by triad promoters, you mean promoters of genes that are present in all three homeologs, and unbalanced means that although the promoters are there, the TFBS are not?

Response: Yes, triad promoters are promoters of genes that are present upstream of triad genes in all subgenomes. Unbalanced binding is defined based on quantitative comparison of TF binding across subgenomes, detailed description is in lines 477-489. The purpose of using quantitative comparison is to reduce the multiple test error.

Line 212. This is very intriguing, and good evidence for functional relevance. This analysis would be enhanced if it were possible to know the age of these TEs. At least in some cases, could the LTRs be used to date these insertions; presumably they would be prior to the divergence of the subgenomes.

Response: We thank the reviewer for the suggestion. To determine the occurrence of RLC_famc1.4 amplification, we collected RLC_famc1.4 from the related Triticeae species including rye and barley. K2P distances reflecting the genetic distance of RLC_famc1.4 between species were calculated. The distribution of K2P distance between wheat subgenomes and wheat-rye shared a peak centered on similar K2P distance, suggesting that a paleo RLC_famc1.4 expansion event occurred prior to the divergence of wheat and rye (following Fig. a). In contrast, for RLG_famc7.3 and RLG_famc13, no common K2P distance peaks were detected between the wheat subgenomes and wheat-rye, suggesting a wheat-specific expansion of these two subfamilies. Analysis of DHS data, which reflects chromatin openness and activity *in vivo*, revealed that RLC_famc1.4 is the top enriched TF family for DHS derived from both TE and dTE (following Fig. b). Thigh enrichment of RLC_famc1.4 in A-B-D homologous TFBS further support the ancient expansion and exaptation of RLC_famc1.4 (following figure c). Please refer to Fig. 4, Fig.7 and lines 258-268.

Line 216. It's hard to imagine how they could have no functional significance (at least historically) if they are conserved.

Response: We agreed with the reviewer's comment and removed the statement of "Despite we do not expect that all these TE domestication events have functional implications".

Figure 4l. This is a bit difficult to understand. The dTE TFBSs would be expected to be more conserved relative to each other than TFBSs that occur "naturally" because even when they are degenerate, the TE sequences flanking the transposon born TFBS would be expected to be more similar than the random sequences flanking the other TFBSs. It's also not clear whether or not the data set is biased, because of the search algorithm would pull out the subset of what may be a very large number of TEs in the same family that does not have conservation of this sequence, which is, after all, quite short.

Response: This conjecture is reasonable. However, we characterized the epigenetic signatures surrounding dTE-derived TFBS, which also exhibited lower activity in the flanking regions (Fig. 6i). We infer that the TE regions are highly active and that the nonfunctional sequences underwent rapid turnover after long-term degeneration. Therefore, sequence conservation may be lower compared to nonTE regions. It also derives from the definition of dTE as a region that partially matches a TE but no longer has a typical TE sequence.

Fig. 6i, Epigenetic profile surrounding dTE-derived and nonTE-derived TFBS. The average levels of DHS, H3K4me3, H3K9ac and DNA methylation were plotted.

Line 228. Because selection has favored this sequence for other reasons in this particular TE?

Response: This is an attractive hypothesis. We performed additional analysis and discussed the possibility why RLG_famc1.4 is selected in the last paragraph of Discussion, including the maintenance of higher-order structure, and the “enhancer runaway” model (lines 318-329).

Line 264. Do we actually know that wheat is particularly “plastic” with respect to transcriptional regulation other than the changes that we would expect in any polyploid? What is the evidence for this?

Response: We realized that the previous statement is not a good summary of this paragraph. In terms of the level of “plasticity”, given the multifaceted effects of TEs on transcriptional regulation, high TE content and diversified TE context in wheat regulatory regions, TEs may have profound implications for wheat transcriptional regulation. However, since there is no direct evidence and comparison with other polyploid is not the main result and conclusion of this manuscript, we have changed the statement to “*Thus, TEs have significantly and continuously rewired the wheat regulatory circuit. Following polyploidization, trans-factors immediately acquired additional suites of cis-elements (10.1038/s41467-019-13386-w), which generated increasingly complex interactions potentially shaped by TEs. This considerable increase in the number of new interactions may have had an immediate or much lagged effect on the adaptation of polyploid wheat (10.1016/j.pbi.2012.03.011, 10.3389/fgene.2020.00792)*” (lines 291-296).

Line 281. On first evolutionary principles, this argument makes very little sense. Any given insertion carrying a random regulatory motif would almost certainly be selected against, regardless of its potential utility. If that is the case, then how would selection

ever favor this kind of speculative “evolvability” function? Are the authors suggesting that selection somehow favored the evolution of TFBSs in the TEs in anticipation of their potential benefits?

Response: We thank the reviewer for pointing this out. The previous description of the recent publication and the relationship to the current result is vague, and we revised it to “*A recent genome-wide characterization of common wheat TEs revealed that despite the intergenic turnover by TEs, an unexpected preservation of the relative distance to genes was observed for specific TE families, implying certain TE families may have been commonly selected in the diploid progenitors during evolution after the divergence from the common ancestor. In accordance with this earlier finding, the present study demonstrated that the position-specific retention of TFBSs in specific TE families occurred in parallel across subgenomes. This apparent sequence conservation of TE-derived TFBS across subgenomes reflected their functional significance.*” (lines 309-317).

Reviewers' Comments:

Reviewer #1:

Remarks to the Author:

I appreciate the efforts made by the authors to improve the manuscript. However, I still find it hard to follow because of a lack of careful use of terminology and evolutionary thinking.

- 1) The abstract still claims a "comprehensive" regulatory network of only 189 TFs.
- 2) The HC, MC and LC datasets on page 3 ought to correspond to the 189 TFs, but $45+48+97 = 190$. This could be clearer.
- 3) I find the concept of "homology" to be abused on page 4. The intergenic regions between pairs of homologous genes are also assumed to be homologous (inherited from the common ancestor) regardless of whether they maintain sequence similarity or not. Homology is not a statement of alignability. The authors are trying to discuss gain/loss of TFBS, but not doing so clearly.
- 4) On line 166, the authors are still not being clear about pre/post polyploidy events, since sub genome and progenitor genome are different ideas.

Reviewer #2:

Remarks to the Author:

The authors addressed most of my previous concerns.

Reviewer #3:

Remarks to the Author:

The authors have done a great deal to allay my concerns, but I still have some nagging issues, mostly having to do with the evidence that the dTEs really are (or were) TEs, and the movement in the text from providing evidence that these TEs may be rewiring regulatory networks (something that I would be really excited by), to the conclusion that they have definitely been exapted. Perhaps I am being overly cautious, but I think there is ample evidence for humans, for instance, that TE involvement in gene regulation is often over-interpreted. With that in mind, I hope the authors are patient with my additional questions. I suppose what I am looking for framing that starts with the null hypothesis and then shows how unlikely it is given the evidence provided. For me, the most striking evidence would be that TE sequences that you know, for sure, were once TEs are syntenically conserved, mediate open chromatin and are associated with specific patterns of expression. For instance, let's take one of the really ancient dTEs. Are there cases where is present in two of the three subgenomes and is associated with a difference in expression? Below are my specific comments.

Figure 8. Again, although I do understand the figure, the wording remains a bit confusing. Since wheat has two diploid progenitors (plural) should it read "progenitors", not progenitor?

Line 62. Perhaps define auto and allopolyploids here.

Line 68. Purely optional, but perhaps a figure here simply outlining the proposed sequence of events giving rise to modern wheat would be helpful to readers not familiar with polyploidy.

Line 88. Again, purely optional, but perhaps it would be helpful to have a paragraph that introduces

the known roles that TEs can play in rewiring regulatory networks.

Figure 1a. Functional, not function, genes.

Line 101. The sentence does not make sense. Given that there might be non-canonical binding, the data were deposited in a data base? So that others can make sense of the data?

Line 108. No gene models in this?

Line 110. Where is the subgenome data in 1c?

Line 176. I'm sure the authors must find this tiresome, but the text still reads as if the conclusion is built into the observation. This in accordance with the fact that most of the wheat genome is composed of repetitive TEs, which are rarely at syntenic positions, so most non-syntenic binding sites, even, or even especially, if they have no functional significance, might be expected to be found in TEs. But at this point, it is not demonstrated that these are in fact "regulatory".

Line 180. So, if you looked at all of the TEs in a given family with single copy reads, I'm assuming that the vast majority of them would not correspond to open chromatin? Again, what I'm suggesting is that it would be helpful to state the null hypothesis clearly here in the results. So, let's say that a particular TE has TFBSs in order to enhance its own (not flanking gene) transcription. If that were the case, then wouldn't this explain these data without invoking a role for this TE in flanking gene expression?

Line 206. Presumably, none (?) of these highly conserved AP2 binding domains were within TEs, whereas many more of the less conserved ones were?

Line 220. Ah, but I see that is not the case.

Line 221. of "both" the balanced and unbalanced?

Line 234. This is an important result if true. Data from a wide variety of plants suggest that sequences that do not have a functional role are rapidly lost via deletion. If these sequences are indeed 1) legitimate TFBS 2) derived from TEs and 3) retained over 5 million years, I would be much more supportive of evidence of more recent exaptation events. In this case, we would have to be convinced that each of these is in fact the case. One concern I have is that once the definition of TEs has been expanded to dTEs, 85% of all the TF binding is in TEs (S6). Is that accurate? And I'm a bit confused by some of the data in S7. bHLH-7D-2 binding sites are almost exclusively in conserved homologous sites that are in TEs. Is that true? Also, I'm probably just misinterpreting the figure, but how can the numbers for the "fraction of TE embedded TFBSs" for homologous and non-homologous sites add up to more than 100%?

Line 235. Perhaps an alignment of a representative set of these balanced TE-derived TFBSs would be helpful, as they could show that are 1) still recognizable TEs and 2) are more conserved in the binding site.

Line 237. "wheat species with different ploidy levels" Are you referring to the subgenomes, or to other extant wheat species?

Figure 6c. Copy number of TEs usually refers to the overall number of TEs. Here, I think you mean the fraction of promoters with TE-derived TFBSs, as if suggested by figure 6e. Also, here and throughout, I think you want "balanced" and "unbalanced".

Figure 6h. This is surprising, as it suggests that that TE-derived binding sites are actually more open and accessible than non-TE derived binding sites. Why would this be the case?

Line 255. was associated with amplification of RLC_famc1.4?

Line 282. Here is where I still hesitate. Do the authors believe that they have conclusively demonstrated exaptation, or that they provide evidence consistent with exaptation?

Line 292. Awkward wording, since there is not just one regulatory circuit in wheat.

Line 293. Do we know this was "immediate"?

Line 299. Now I'm confused. Where was the evidence for convergent, as opposed to parallel, evolution? Are you saying that different TE insertions ended up having the same effect on gene expression?

Line 302. Wording. Regulatory networks, not the regulatory network.

Line 305. This bit would work well in the introduction.

Line 312. Or that TEs differ in their targeting? MULE elements, both ancient and recent, are found close to TSSs due to targeting, not selection.

Line 319. In a con way?

Line 321. How is this 'enhancer runaway', which implies a positive feedback loop.

Line 522. It still isn't clear how dTEs were defined. This is key to the hypothesis, but how confident are you that these dTEs are in fact derived from TE sequences?

Reviewer #1 (Remarks to the Author):

I appreciate the efforts made by the authors to improve the manuscript. However, I still find it hard to follow because of a lack of careful use of terminology and evolutionary thinking.

1) The abstract still claims a "comprehensive" regulatory network of only 189 TFs.

Response: 'comprehensive' is now removed.

2) The HC, MC and LC datasets on page 3 ought to correspond to the 189 TFs, but $45+48+97=190$. This could be clearer.

Response: We thank the reviewer for pointing this out. There was one extra duplicate in the dataset, which we removed, leaving 189 datasets in the revision.

3) I find the concept of "homology" to be abused on page 4. The intergenic regions between pairs of homologous genes are also assumed to be homologous (inherited from the common ancestor) regardless of whether they maintain sequence similarity or not. Homology is not a statement of alignability. The authors are trying to discuss gain/loss of TFBS, but not doing so clearly.

Response: We thank the reviewer for pointing this out. We changed "subgenome-homologous" and "subgenome-nonhomologous" to subgenome-homoeologous (i.e., homologous and syntenic) and subgenome-specific (i.e., unalignable), respectively. Please refer to lines 154-158, 162,463-465, 636,660 and labels in Figure 2-4.

4) On line 166, the authors are still not being clear about pre/post polyploidy events, since sub genome and progenitor genome are different ideas.

Response: We thank the reviewer for pointing this out. We revised "expansion event in subgenome" to "expansion event during evolution" (line 176).

Reviewer #2 (Remarks to the Author):

The authors addressed most of my previous concerns.

Reviewer #3 (Remarks to the Author):

The authors have done a great deal to allay my concerns, but I still have some nagging issues, mostly having to do with the evidence that the dTEs really are (or were) TEs, and the movement in the text from providing evidence that these TEs may be rewiring regulatory networks (something that I would be really excited by), to the conclusion that they have definitely been exapted. Perhaps I am being overly cautious, but I think there is ample evidence for humans, for instance, that TE involvement in gene regulation is often over-interpreted. With that in mind, I hope the authors are patient with my additional questions. I suppose what I am looking for framing that starts with

the null hypothesis and then shows how unlikely it is given the evidence provided. For me, the most striking evidence would be that TE sequences that you know, for sure, were once TEs are syntenically conserved, mediate open chromatin and are associated with specific patterns of expression. For instance, let's take one of the really ancient dTEs. Are there cases where is present in two of the three subgenomes and is associated with a difference in expression?

Response: we understand the reviewer's concern about the true functionality of these (d)TE-derived TFBS, and we appreciate the insightful comments. Accordingly, we included statistical tests for dTE definition, and added the analysis regarding the relationship with open chromatin and expression.

Firstly, dTE is defined based on both synteny (triad promoter) and sequence similarity. we added a detailed description (lines 245-248) and schematic diagram to illustrate how dTE is defined (Fig.6d). For each triad gene (i.e., 1:1:1 correspondence across subgenomes) with at least one gene containing TE-embedded TFBS in the promoter, regions in the other one or two triad promoters alignable to the TE-embedded TFBS were defined as dTEs. The dTE-embedded TFBS was highly similar to the corresponding TE-embedded TFBS (Fig. 6f), suggesting a common origin.

Secondly, as a control, the TEs contributed to TFBS from one triad promoter was compared to the promoters of different triads without typical TE structure. For each TF, 1000 permutation test was conducted, significantly fewer alignable region was detected using the same cutoff as above (lines 252-257 and Fig. S10). The definition of dTE is now more complete from a statistical point of view.

Thirdly, we detected the linkage between (d)TE-derived balanced binding and balanced expression. We found that when TEs and dTEs were considered together, the fraction of balanced TFBS derived from (d)TEs within triads increased substantially for all three triad members (Fig. 6e, orange box). We further compared the density of (d)TE-derived DHS across triad promoters and gene expression, and detected clear association between balanced binding and balanced expression (Fig. 6k-l).

Below are my specific comments.

1. Figure 8. Again, although I do understand the figure, the wording remains a bit confusing. Since wheat has two diploid progenitors (plural) should it read “progenitors”, not progenitor?

Response: We thank the reviewer for pointing this out. ‘progenitor’ is now changed to ‘progenitors’.

2. Line 62. Perhaps define auto and allopolyploids here.

Response: The auto- and allo-polyploids are defined in lines 61-62.

3. Line 68. Purely optional, but perhaps a figure here simply outlining the proposed sequence of events giving rise to modern wheat would be helpful to readers not familiar with polyploidy.

Response: We agree with the reviewer’s suggestion that putting a figure here facilitates the understanding of readers. Please refer to Figure 1a of the revised manuscript.

4. Line 88. Again, purely optional, but perhaps it would be helpful to have a paragraph that introduces the known roles that TEs can play in rewiring regulatory networks.

Response: We thank the reviewer for pointing this out. We added the description about the known roles of TEs in rewiring regulatory network in plants and animals (lines 84-85, lines 89-93).

5. Figure 1a. Functional, not function, genes.

Response: Corrected.

6. Line 101. The sentence does not make sense. Given that there might be non-canonical binding, the data were deposited in a data base? So that others can make sense of the data?

Response: We revised the sentence to “All DAP-seq data and peak files were deposited in GEO database (<https://www.ncbi.nlm.nih.gov/geo/GSE192815>; reviewer token: appgkssuzvabdm). The HC and MC TFs were used for the subsequent analysis.” (line 109-112).

7. Line 108. No gene models in this?

Response: Gene track is in the orange box. We revised the title to make it clearer.

8. Line 110. Where is the subgenome data in 1c?

Response: Subgenome information is included in the enlarged heatmap in the upper right corner. TF names ending in A, B, and D indicate which subgenome they belong to. The binding of homeologous TFs was largely similar across subgenomes.

9. Line 176. I’m sure the authors must find this tiresome, but the text still reads as if the conclusion is built into the observation. This in accordance with the fact that most of the wheat genome is composed of repetitive TEs, which are rarely at syntenic positions, so most non-syntenic binding sites, even, or even especially, if they have no functional significance, might be expected to be found in TEs. But at this point, it is not demonstrated that these are in fact “regulatory”.

Response: We agree with the reviewer, and included the alternative possibilities and evolutionary implications of TE-embedded TFBS in lines 191-200.

10. Line 180. So, if you looked at all of the TEs in a given family with single copy reads, I’m assuming that the vast majority of them would not correspond to open chromatin? Again, what I’m suggesting is that it would be helpful to state the null hypothesis clearly here in the results. So, let’s say that a particular TE has TFBSs in order to enhance its own (not flanking gene) transcription. If that were the case, then wouldn’t this explain these data without invoking a role for this TE in flanking gene expression?

Response: Please refer to the above response. The alternative explanation and evolutionary implications are added to lines 191-200.

11. Line 206. Presumably, none (?) of these highly conserved AP2 binding domains were within TEs, whereas many more of the less conserved ones were?

Response: Line 206 (line 223 in the current version) describes the result in Figure 5, which focuses on the quantitative comparison of the binding densities in promoters of triad genes (1:1:1 correspondence across subgenomes). The ratios of TEs are in Figure 4a. There is a small fraction of AP2 binding within TEs (8% - 12%). We reduced the radius of the circle to make the scale display more accurate.

12. Line 220. Ah, but I see that is not the case.

Response: We revised the sentence to “Balanced and unbalanced TFBSs largely have similar fractions overlapping with TEs, with slightly higher ratios for balanced TFBSs (Fig. 6a–b).” (lines 236-238).

13. Line 221. of “both” the balanced and unbalanced?

Response: Revised to “both” balanced and unbalanced TF binding (line 239).

14. Line 234. This is an important result if true. Data from a wide variety of plants suggest that sequences that do not have a functional role are rapidly lost via deletion. If these sequences are indeed 1) legitimate TFBS 2) derived from TEs and 3) retained over 5 million years, I would be much more supportive of evidence of more recent exaptation events. In this case, we would have to be convinced that each of these is in fact the case. One concern I have is that once the definition of TEs has been expanded to dTEs, 85% of all the TF binding is in TEs (S6). Is that accurate? And I’m a bit confused by some of the data in S7. bHLH-7D-2 binding sites are almost exclusively in conserved homologous sites that are in TEs. Is that true? Also, I’m probably just misinterpreting the figure, but how can the numbers for the “fraction of TE embedded TFBSs” for homologous and non-homologous sites add up to more than 100%?

Response:

1) legitimate TFBS. We included DHS data to reflect TFBS activity *in vivo*. 22% - 35% expanded DHS-TFBS are embedded in TEs, whereas only 11% - 13% non-expanded DHS-TFBS in TEs, indicating that expansion of *in vivo* active TFBS within subgenomes is likely associated with TE expansion (Fig. S5, bottom panel).

2) derived from TEs. The TEs here are defined based on the standard pipeline as previously described with canonical TE structure detected (Methods, 480-487).

3) retained over 5 million years. We included multiple evidence demonstrating the conservation of TE-TFBS and dTEs. i) Significantly higher conservation of TE-embedded TFBS loci as compared to random TE regions were detected. Conservation score is calculated by comparing across wheat species with different levels of polyploidy (Fig. S7). ii) An apparent expansion event of RLC_famc1.4 was shared between wheat and barley, indicating an ancestral expansion of RLC_famc1.4. RLC_famc 1.4 is the top enriched TE family contributing to TFBS conserved across subgenomes, providing further support about the impact of ancient TE expansion on subgenome conserved TFBS (Fig.7). iii) Permutation tests suggested the high statistical significance of dTEs detected (Fig. S10).

Figure S6 (Figure S5 in revised version) indicated the fraction of expanded TFBS within annotated TEs. The definition of dTE was not introduced here, which was proposed in Figure 6 when considering the subgenome-balanced and unbalanced bindings in gene promoters.

Figure S7 (Figure S6 in revised version). The odd ratio is due to the low number of TFBS. We now delete TFs with subgenome homoeologous or specific TFBS numbers less than 100.

15. Line 235. Perhaps an alignment of a representative set of these balanced TE-derived TFBS would be helpful, as they could show that are 1) still recognizable TEs and 2) are more conserved in the binding site.

Response: We agree with the reviewer’s suggestion and added the figure of alignment to Fig. 6g-i.

16. Line 237. “wheat species with different ploidy levels” Are you referring to the subgenomes, or to other extant wheat species?

Response: It refers to diploid and tetraploid progenitors. We now include this information (lines 262-263).

17. Figure 6c. Copy number of TEs usually refers to the overall number of TEs. Here, I think you mean the fraction of promoters with TE-derived TFBSs, as if suggested by figure 6e. Also, here and throughout, I think you want “balanced” and “unbalanced”.

Response: The previous version is misleading. We revised the legends of Fig. 6c and 6e according to the suggestion.

18. Figure 6h. This is surprising, as it suggests that that TE-derived binding sites are actually more open and accessible than non-TE derived binding sites. Why would this be the case?

Response: To avoid bias in the average profile caused by outliers, we calculated the distribution of epigenetic features and conservation score surrounding dTE-derived TFBS, which is consistent with the results of profiles. We postulated that this may derived from the definition of dTEs (lines 245-248), which are TE remnants but no longer has typical TE structure. Thus, these dTE-derived TFBS potentially under relatively strong purifying selection, and showed higher conservation and activity.

19. Line 255. was associated with amplification of RLC_famc1.4?

Response: Corrected as suggested (line 280).

20. Line 282. Here is where I still hesitate. Do the authors believe that they have conclusively demonstrated exaptation, or that they provide evidence consistent with exaptation?

Response: We revised “TE exaptation” to “TEs contributed to TFBS” (line 309).

21. Line 292. Awkward wording, since there is not just one regulatory circuit in wheat.

Response: “the regulatory circuit” is revised to “regulatory circuits” (line 317).

22. Line 293. Do we know this was “immediate”?

Response: ‘immediate’ is now removed.

23. Line 299. Now I'm confused. Where was the evidence for convergent, as opposed to parallel, evolution? Are you saying that different TE insertions ended up having the same effect on gene expression?

Response: We agree with the reviewer's concern. Using "parallel evolution" is appropriate (line 324).

24. Line 302. Wording. Regulatory networks, not the regulatory network.

Response: corrected (line 327).

25. Line 305. This bit would work well in the introduction.

Response: We added description of the known roles of TEs in rewiring regulatory network to the Introduction (lines 84-86, lines 89-93).

26. Line 312. Or that TEs differ in their targeting? MULE elements, both ancient and recent, are found close to TSSs due to targeting, not selection.

Response: We thank the reviewer for the information, which we added to the Discussion "implying certain TE families may have insertion preference relative to genes(10.1038/nature02953)" (line 337).

27. Line 319. In a con way?

Response: It is corrected to "conserved" (line 343).

28. Line 321. How is this 'enhancer runaway', which implies a positive feedback loop.

Response: This explanation is somewhat unreasonable. We revised this part of the Discussion (lines 343-347).

29. Line 522. It still isn't clear how dTEs were defined. This is key to the hypothesis, but how confident are you that these dTEs are in fact derived from TE sequences?

Response: Please refer to the response to the major comment, we included detailed description of dTE-TFBS definition and added statistical test (lines 245-257, Fig. 6 and Fig. S10).

Reviewers' Comments:

Reviewer #1:

Remarks to the Author:

The authors appear to have addressed my prior concerns, and I have no further comments.

Reviewer #3:

Remarks to the Author:

The authors have done an excellent job at responding to my concerns. I hope they agree that the result has been an improved manuscript and look forward to seeing it in print.

REVIEWERS' COMMENTS

Reviewer #1 (Remarks to the Author):

The authors appear to have addressed my prior concerns, and I have no further comments.

Response: We thank the reviewer for the supportive and constructive comments in helping us improve our manuscript.

Reviewer #3 (Remarks to the Author):

The authors have done an excellent job at responding to my concerns. I hope they agree that the result has been an improved manuscript and look forward to seeing it in print.

Response: We thank the reviewer for the insightful advice and the patience in helping us improve our manuscript.